# LARGE-SCALE MOLECULAR DYNAMICS SIMULATION: DIRECT INTERATOMIC MODELING WITH DILATED MESSAGE PASSING

## ABSTRACT

Large-scale molecular dynamics simulation is essential in understanding chemical and biological processes, necessitating the accurate and efficient modeling of interatomic interactions. Existing learning-based methods generally are based on message passing mechanisms; they either are not scalable or are too coarse to offer accurate modeling. We propose a new message passing framework that can effectively and efficiently model interatomic interactions for simulating large-scale molecular dynamics at full atomic resolution. Specifically, our framework is stacked with a sequence of message passing neural network layers, each realizing the message passing over a distinct and dilated star-structured path. These star-structured paths are constructed progressively along dilated regions to capture the distance-dependent interactions. The crux of our framework is that it resolves the problem of dense interatomic interactions of large-scale atomic systems with sparser and region-based message passing graphs. We evaluate the framework on four benchmarks: the MD22 (molecules with 42–370 atoms), the Chignolin (a 166-atom protein featuring diverse conformations), the AdK dataset (a protein trajectory with up to 3,000 atoms), and the MISATO dataset (over 10,000 heterogeneous protein-ligand complexes, including systems with up to 40,000 atoms). Comprehensive evaluations demonstrate that our approach delivers state-of-the-art performance overall across various benchmarks. In particular, it is the first learning-based method to achieve atomic-level accuracy in protein-ligand dynamics simulation while preserving computational efficiency.

## 1 INTRODUCTION

The simulation of large-scale molecular dynamics systems, such as protein-ligand binding, is crucial for understanding biological processes and advancing drug discovery (Yasuda et al., 2022; Yang et al., 2020; Lahey & Rowley, 2020). Machine learning (ML) has emerged as a promising paradigm, attempting to efficiently and accurately simulate molecular dynamics (Wang et al., 2024a). Existing ML methods typically represent molecules using graphs with a predefined radius cutoff (Liao et al., 2024; Wang et al., 2024c). With molecular graph modeling, various equivariant graph neural networks (EGNNs) are proposed to capture the geometric structures of molecular systems by incorporating the inductive bias of symmetry (Satorras et al., 2021; Han et al., 2022). Existing EGNNs generally fall into the message passing framework, which models the interatomic interactions via local message passing. They have been shown promising performance for small molecular systems (e.g., MD22 (Chmiela et al., 2023)), illustrating their capability to capture local environments.

For simulating large molecular systems (e.g., chignolin (Wang et al., 2023a), AdK (Seyler & Beckstein, 2017)), a straightforward strategy is to increase the number of message passing neural network (MPNN) layers or enlarge the radius cutoff. Nevertheless, increasing the number of MPNN layers inevitably introduces the phenomena of over-squashing and over-smoothing (Alon & Yahav, 2021; Gutteridge et al., 2023), degrading the geometric expressiveness (see Figure 1 (a)); A larger radius cutoff would lead to denser computational graphs, incurring larger computation overhead (see Figure 1 (b)). To efficiently and effectively simulate the large-scale molecule systems, several works attempt to reduce the scale of the message passing computational graph based on fragmentation (Wang et al., 2024a;c; Unke et al., 2024) or solely modeling the backbone atoms (Wu et al.,

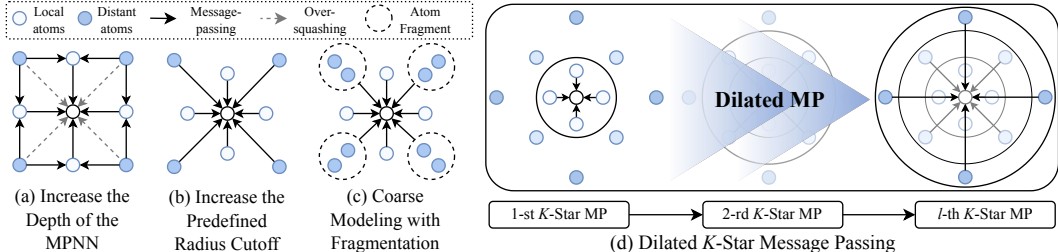

Figure 1: *Illustration of Existing Methods for Large-Scale MD and the Proposed Dilated K-Star Message Passing.* (a) Stacking layers leads to indirect interaction modeling and over-smoothing/over-squashing (b) GNNs with increased radius cutoff models incur high computational costs with dense computational graphs. (c) Coarse modeling can improve efficiency but incurs sub-optimal accuracy when modeling complexes with heterogeneous structures. (d) Our approach models dense interatomic interactions with a sequence of MPNN, each achieving the message passing over a distinct and dilated star-structured path.

2024; Han et al., 2022; Liu et al., 2024). However, they focus on homogeneous complexes and fail to model heterogeneous ones like protein-ligand systems. Additionally, they yield sub-optimal accuracy due to the coarse-grained sparsification (see Figure 1 (c)). Therefore, fine-granularity modeling, *a.k.a* full-atomic modeling, is desirable for effectively simulating large-scale complexes.

In this paper, we propose **D**ilated K-star **M**essage-**P**assing, DKMP*, a new framework that can capture the pairwise atomic interactions for effectively and efficiently simulating large-scale molecular dynamics at a full-atomic level. Specifically, DKMP* models pairwise atom interactions regionally by stacking a sequence of one-layer MPNNs, propagating messages within a set of K-star message passing graphs for each atom, where K denotes the number of neighbors (see Figure 1 (d)). These K-star graphs are constructed with region-based sampling mechanisms reflecting the distance-dependent interaction nature of force fields. This allows for preserving the sparsity of the geometric graph for effective and efficient modeling, and simultaneously alleviating over-squashing. Additionally, our sequential and dilated message-passing (MP) would prevent distant atoms from interacting at earlier iterations, respecting the locality of the graph. More specifically, the **effectiveness** of our DKMP* lies in the following two aspects: *1)* Conceptually, DKMP* is grounded in the physical fact that pairwise potentials are governed by the distance-dependent formula (Bartók et al., 2010; Tkatchenko & Scheffler, 2009), e.g., the intensity of interatomic interactions is proportional to a power of the distance. *2)* Technically, the $K$-star message passing essentially falls into the graph rewiring framework (Gutteridge et al., 2023; Attali et al., 2024) that addresses the over-squashing in large-scale molecular systems. DKMP* is **efficient** as it models instant interatomic interactions via dilated and sparser computational graphs. We theoretically prove that DKMP* avoids over-squashing and analyze its computational efficiency advantage.

We instantiated our design with two implementations for different MD scenarios and evaluated them over a series of standard benchmarks. *First*, we implemented the $K$-star message passing graph construction by expanding radius cutoff intervals and evaluated on MD22 (Chmiela et al., 2023) and Chignolin (Wang et al., 2023a) datasets, respectively. We observe that our method achieves state-of-the-art accuracy and efficiency as the size of the molecular system scales up. *Second*, we consider a larger-scale and more challenging scenario involving structurally heterogeneous bio-complexes—the MISATO dataset (Siebenmorgen et al., 2024); this dataset includes variable scales of protein-ligand complexes ranging from 556 to 40,798 atoms. As the molecular size varies significantly, dilating radius cutoffs would introduce comprehensive computation costs. Therefore, we further craft a dilation mechanism based on distance ranking equipped with dilated graph attention to accelerate the message passing process. The experimental results show that it can reduce modeling errors by up to three orders of magnitude and achieve atomic-level accuracy in per-atom dynamics simulations of protein-ligand complexes containing up to 40,000 atoms.

## 2 PRELIMINARIES

**Molecular Graph**. In this paper, we explore the dynamics simulation of large-scale molecular systems, represented as a sequence of geometric graphs $\mathcal{G}^t$ indexed by time $t$. Suppose we have $N$ atoms in the system, then the molecular system $\mathcal{G}^t$ at each snapshot can be represented as a point

cloud denoted as $\mathcal{G}^t = \langle X^t, H \rangle$, where $X^t = [\mathbf{x}_1^t; \ldots; \mathbf{x}_N^t] \in \mathbb{R}^{N \times 3}$ is the atom coordinate matrix and $H = [\mathbf{h}_1; \ldots; \mathbf{h}_N] \in \mathbb{R}^{N \times h}$ is the node feature matrix. $H$ typically contains atomic types or charge features, and it is generally time-invariant.

Given the molecular structure $\mathcal{G}^t$, the objective of dynamics simulation is to predict the molecule structure at time $t + 1$ or to predict the energy or force at time $t$ to update the dynamics. Specifically, the future coordinates $X^{t+1}$ are either directly estimated by $X^{t+1} = \phi_\theta(\mathcal{G}^t)$ or indirectly estimated through node-level forces $\mathcal{F}^t \in \mathbb{R}^{N \times 3} = \vartheta_\theta(\mathcal{G}^t)$ or the graph-level system energy $U^t \in \mathbb{R} = \varphi_\theta(\mathcal{G}^t)$. These estimated quantities are then used to update the molecular dynamics by solving the differential equations that describe the system's behavior. Machine learning-based approaches for MD simulation are broadly categorized into two paradigms: Structure-to-Structure (S2S), which directly estimates future configurations, and Structure-to-Energy-and-Forces (S2EF), which computes energy/force quantities. An illustration of these two tasks is presented in Figure 2. We briefly review machine learning-based molecular dynamics simulations in Appendix A.

**Message Passing Flow in a Molecular Graph**. Given a molecular graph, $\mathcal{G}$, mainstream methods for molecular dynamics (MD) employ a predefined radius cutoff $C$ to construct the message passing graph $G = (V, E)$. Here, $V = [X; H] \in \mathbb{R}^{N \times (3+h)}$ represents node features concatenating spatial coordinates $X$ and atomic attributes $H$, while $E$ comprises virtual edges connecting all node pairs within Euclidean distance $C$. The graph structure is represented by an adjacency matrix $\mathbf{A} \in \mathbb{R}^{N \times N}$, where each entry $\mathbf{A}(i, j)$ equals 1 if the geometric distance between nodes $i$ and $j$, denoted by $\mathrm{dist}(i, j)$, satisfies $\mathrm{dist}(i, j) \leq C$. For a node $i$, this adjacency matrix partitions the graph into level sets based on the shortest-path distance $d_G(i, j) : V \times V \to \mathbb{R}_{\geq 0}$, defining the $h$-hop neighborhood:

$$\mathcal{N}_h^H(i) := \{j \in V : d_G(i, j) = h\}, \tag{1}$$

where $d_G(i, j)$ denotes the minimal path length between nodes $i$ and $j$. The 1-hop neighborhood $\mathcal{N}_1^H(i)$ corresponds to atoms within the radius cutoff $C$. Crucially, the $h$-hop neighborhood includes atoms at exact distance $h$. Accordingly, for an atom $i$ to exchange information with atom $j \in \mathcal{N}_k^H(i)$, existing methods require either a minimum of $h$ layers or an $h$-fold increase in the radius cutoff.

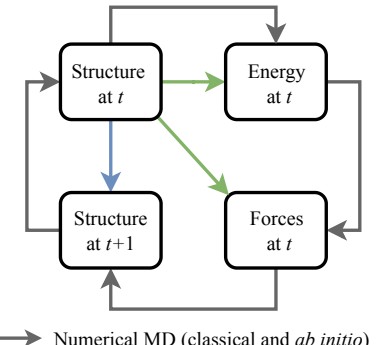

Figure 2: *Illustration of Molecule Dynamics Simulation Methods*: Black lines denote traditional MD simulation methods. Green lines denote the task of structure to energy and forces (S2EF). Blue line denotes the task of structure to structure (S2S).

# 3 OUR FRAMEWORK: DKMP*

## 3.1 MESSAGE PASSING OVER DILATED K-STAR GRAPHS

To model dense interatomic interactions for large-scale MD simulations without compromising on full atoms, we propose our novel dilated $K$-star message passing framework. The idea of our work is inspired by the physical fact that the intensity of pairwise interatomic potentials varies according to the distances, including covalent bonds, ionic interactions, van der Waals forces, and the Lennard-Jones potential (Bartók et al., 2010; Tkatchenko & Scheffler, 2009).

$K$-**Star Message Passing**. To contextualize our framework, we consider two atoms $i, j \in V$ separated by a distance $r$ ($r/C > 1$). Conventional MPNNs would require $\lceil r/C \rceil$ layers or a $r/C$-fold increase in radius cutoff to enable such interactions. Drawing on physical intuition, we argue that distant atom pairs should engage in direct interactions rather than relying solely on iterative neighbor-mediated communication. Furthermore, since the expansion of the radius cutoff incurs prohibitive computational costs, we propose stacking $L$ MPNNs, each implementing a distinct $K$-star message passing scheme. Specifically, for the $l$-th MPNN ($l \in \{1, \ldots, L\}$), we construct a dedicated message passing graph $G_l = (V, E_l)$ featuring dilated connectivity, where $E_l$ contains virtual edges connecting atom pairs through progressively growing receptive fields.

**The Dilation Mechanism**. $K$-star message passing ensures a direct message passing between atomic pairs and avoids the issue of over-squashing. Next, we elaborate on the dilation mechanism that progressively expands the receptive fields of $K$-star MPNNs. We define our dilated MPNNs through four structural constraints on $E_1, \ldots, E_L$:

$$
\begin{aligned}
(i) \quad &\textbf{Mutual-exclusion}: &&E_i \cap E_j = \emptyset \,\forall i \neq j, \\
(ii) \quad &\textbf{Monotonicity}: &&\forall e \in E_i, e' \in E_{i+1}, d(e) \leq d(e'), \\
(iii) \quad &\textbf{Non-empty}: &&\forall l \in \{1, \ldots, L\}, E_l \neq \emptyset, \\
(iv) \quad &\textbf{Completeness}: &&\exists \hat{L} \in \mathbb{N} : \bigcup_{i=1}^{\hat{L}} E_i = \hat{E},
\end{aligned}
\tag{2}
$$

where $\hat{E}$ denotes the complete edge set of graph $G$ under full atomic connectivity, and $d(\cdot)$ represents Euclidean edge distance. By definition, DKMP* can be regarded as a contiguous partition of $\hat{E}$ that is sorted in non-decreasing order while retaining the first $L$ subsets. Based on the above definition, atom $i$ in the $l$-th MPNN will communicate only with atoms in $\mathcal{N}_l^K(i) := \{j \in V : e(i,j) \in E_l\}$, i.e., a $K$-star message passing and $K$ is the cardinality of $\mathcal{N}_l^K(i)$.

**Efficient Large-scale Interatomic Interactions Modeling without Over-Squashing**. In this section, we theoretically demonstrate how the DKMP* framework circumvents over-squashing while preserving computational efficiency in large-scale MD simulations. We formalize this property by the following proposition:

**Proposition 3.1** *DKMP\* with the dilation mechanism defined in Eq. 2 alleviates over-squashing in message passing interactions for any two nodes $u, v \in V$ connected by edge $e \in E_l$.*

*Proof.* Consider two atoms $i, j \in V$ separated by distance $r$. In traditional geometric MPNNs with a radius cutoff $C$, atoms $i$ and $j$ first exchange information at layer $c := \lceil r/C \rceil$. Over-smoothing occurs when node features become indistinguishable as the number of layers increases (Oono & Suzuki, 2020). The over-squashing of information occurs when the representation of one node $i$ fails to be affected by some input feature of the node at a distance $r$ from node $i$. By contrast, our method enables direct information exchange at layer $l$ where $e(i,j) \in E_l$, circumventing exponential operations on $\mathbf{A}$ and a direct modeling between nodes with distance $r$, thereby avoiding over-squashing. We give a more formal statement of over-squashing and a proof of Proposition 3.1 in Appendix C.

*Complexity Analysis.* Traditional MPNNs require $\mathcal{O}(L \cdot |E|)$ memory for $L$ layers, while DKMP* achieves $\mathcal{O}\left(\sum_{l=1}^{L} |E_l|\right)$. As $\sum_{l=1}^{L} |E_l| \leq L \cdot |E|$ (with equality only if $L = 1$), DKMP* maintains comparable efficiency to conventional approaches while providing effective expressivity in modeling large-scale interatomic interactions.

### 3.2 Implementing with Different Dilating Strategies

Current learning methods typically are evaluated over two types of datasets involving single-scale (MD22, AdK, and Chignolin) and various-scale (MISATO) molecular systems, respectively. Without loss of generality, we will evaluate these two types of datasets to verify the effectiveness of our framework. A critical observation is that single-scale and various-scale molecular systems exhibit distinct influences for computational efficiency. Specifically, in single-scale systems, the message passing graph remains largely static across simulation steps. In contrast, various-scale systems employing predefined radius cutoffs experience significant fluctuations in the number of interacting atom pairs ($K$), leading to reduced computational efficiency. In this regard, for different dataset scenarios, we implement our DKMP* with various dilating strategies.

#### 3.2.1 Dilating Radius Cutoff Interval

For single-scale molecular systems, we introduce progressive dilated message passing through iterative expansion of radial cutoff intervals (Figure 3 (a)). We initially establish a maximum interatomic interaction distance, designated as the radius cutoff parameter $\mathcal{C}$. For each atom $i \in V$, the coordinates are utilized as the focal point to define two concentric spheres with radius $l\frac{\mathcal{C}}{L}$ and $(l+1)\frac{\mathcal{C}}{L}$,

respectively. Here, $l \in [1, \ldots, L]$ signifies the index of MPNNs. Atoms positioned within these concentric spheres are identified as neighbors of $i$, denoted as $j$. Consequently, within the $l$-th layer of the network, the neighbors of $i$ consist of atoms located at distances from $i$ ranging between $(l-1)\frac{C}{L}$ and $l\frac{C}{L}$, denoted as $\mathcal{N}_l^C(i)$ and expressed as:

$$\mathcal{N}_l^C(i) := \{ j \mid (l-1)\frac{C}{L} \leq \text{dist}(i,j) < l\frac{C}{L} \}. \tag{3}$$

Then, MPNNs dilated by radius cut-off intervals can be denoted as $\{ G_l = (V, E_l^C) \}_1^L$, where $E_l^C := \{ e \mid (l-1)\frac{C}{L} \leq d(e) < l\frac{C}{L} \}$.

Dilating radius cutoff interval enables DKMP* to capture dense interatomic interactions with expanding edge connections. Most existing message passing methods often rely on static graph modeling for updating edge features—limiting their adaptability to our dilated $K$-star graph (Wang et al., 2024c; Liao et al., 2024). To address this, we develop a message passing mechanism designed to accommodate dilated information exchange for our dilated radius cut-off interval implementation.

Figure 3: *Illustration of Two Implementations*: (a) An example of dilated radius cutoff interval. (b) An example of dilated distance-ranking (with fixed $K = 2$).

**Message Passing for Dilating Radius Cutoff.** Our framework implements distinct processing pathways for scalar and vector messages while replacing edge feature updates with implicit edge information aggregation into node features. *1)* For each atom $i$ in layer $l$, we separate scalar node features $m_i^l$ (representing atomic numbers) from vector features $\mathbf{m}_i^l$ (encoding atomic coordinates), following established practices in (Liao et al., 2024; Wang et al., 2024c). *2)* Scalar interactions between nodes $i$ and $j$ are computed using their respective scalar features $\mathbf{z}_i$ and $\mathbf{z}_j$. *3)* Vector message construction between the pair incorporates three components: the scalar message $m_{ij}^l$, the vector state $\mathbf{v}_j^l$ of node $j$, and the directional edge vectors $\mathbf{r}_{ij}$. *4)* After performing neighbor aggregation on both scalar and vector messages for atom $i$, we feed these aggregated signals alongside vector features $\mathbf{v}_i^l$ into a neural network to generate the updated vector message. We provide full architectural details in Appendix D.

### 3.2.2 DILATING DISTANCE RANKING

Handling various-scale datasets through dilated radial cutoff intervals can introduce substantial variance in message passing graphs, resulting in prohibitive computational overhead. To mitigate this, we propose a dilated distance ranking strategy for fixed $K$-star topology message passing (Figure 3 (b)). Our approach first defines $\mathcal{M}(\mathcal{M} \equiv 0 \pmod{L})$, the maximum number of neighbors processed by our DKMP*. For the first MPNN layer, the neighbors of atom $i$ are its $\frac{\mathcal{M}}{L}$ nearest atoms. In subsequent layers, neighbors will be iteratively expanded: the $l$-th layer selects the $l\frac{\mathcal{M}}{L}$ closest atoms while excluding the $(l-1)\frac{\mathcal{M}}{L}$ atoms already modeled in prior layers. This ensures uniform neighbor set sizes across layers. Formally, the neighborhood $\mathcal{N}_l^R(i)$ for the $l$-th layer is defined as:

$$\mathcal{N}_l^R(i) = \left\{ j \mid (l-1)\frac{\mathcal{M}}{L} < \text{rank}(\text{dist}(i,j)) \leq l\frac{\mathcal{M}}{L} \right\}, \tag{4}$$

where $\text{rank}(\text{dist}(i,j))$ denotes the order index of atom $j$ among all neighbors of atom $i$ sorted by ascending distance ranking. Our implementation by dilating distance ranking is structured as $\{ G_l = (V, E_l^R) \}_1^L$, where $E_l^R := \{ e \mid (l-1)\frac{\mathcal{M}}{L} \leq \text{rank}(d(e)) < l\frac{\mathcal{M}}{L} \}$.

**Message Passing for Dilating Distance Ranking.** Our dilating distance ranking mechanism, tailored for various-scale molecular systems, guarantees that each atom consistently aggregates information from a fixed number of neighbors per MPNN. This property enables inherently parallel message aggregation. To further optimize computational performance, we design a dilated graph

attention module specifically within the dilating distance ranking model instance. Drawing on architectural principles from *AlphaFold3* (Abramson et al., 2024) for large-scale protein modeling, we omit equivariance constraints to simplify the architecture, enhancing computational efficiency.

Given the geometric data $\mathcal{G}^t = \langle X^t, H \rangle$ representing a molecule at time $t$ [1], we embed the atomic coordinates $X$ and atomic number $H$ to $\hat{X} = [\mathbf{x}_1; \ldots; \mathbf{x}_N] \in \mathbb{R}^{N \times d}$ and $S = [\mathbf{s}_1; \ldots; \mathbf{s}_N] \in \mathbb{R}^{N \times d}$, respectively. Our dilated graph attention separates the scalar and vector attentions and updates vectors in each layer only as MD simulation only varies the coordinates. The idea is divided into dilated graph scalar cross-attention and dilated graph vector self-attention.

We compute the linear projection vector query ($\mathbf{Q^x}$) from $\hat{X}$, along with scalar and vector keys ($\mathbf{K^s}$, $\mathbf{K^x}$) from $S$ for each atom. The scalar and vector neighborhood attention weights for the $i$-th atom in layer $l$ ($\mathbf{A}_i^{\mathbf{s}}$ and $\mathbf{A}_i^{\mathbf{x}}$) are defined as pairwise inner products between the atom's vector query projection and its $K$ nearest neighbors' scalar/vector key projections. Given neighborhood $\mathcal{N}_l^{R}(i) = \{j_1, j_2, \ldots, j_K\}$ from Eq. (4), the attention weights are formulated as $\mathbf{A}_i^{\mathbf{f}} = [\mathbf{q}_i^{\mathbf{x}} \mathbf{k}_{j_1}^{\mathbf{f}}{}^{\top} + \mathrm{dist}(i, j_1); \mathbf{q}_i^{\mathbf{x}} \mathbf{k}_{j_2}^{\mathbf{f}}{}^{\top} + \mathrm{dist}(i, j_2); \cdots; \mathbf{q}_i^{\mathbf{x}} \mathbf{k}_{j_K}^{\mathbf{f}}{}^{\top} + \mathrm{dist}(i, j_K)]$, $\mathbf{f} = \{\mathbf{s}, \mathbf{x}\}$. The $K$-neighbor vector value matrix $\mathbf{V}_i^{\mathbf{x}}$ aggregates projected features from neighboring atoms: $\mathbf{V}_i^{\mathbf{x}} = [\mathbf{v}_{j_1}^{\top}, \ldots, \mathbf{v}_{j_K}^{\top}]^{\top}$, where $\mathbf{v}_j$ is a linear projection of $i$-th atom's $j$-th neighbor. Dilated graph scalar cross-attention for the $i$-th atom is then defined as $\mathrm{DGA}_{\mathbf{s}}(i) = \mathrm{softmax}(\frac{\mathbf{A}_i^{\mathbf{s}}}{\sqrt{d}}) \mathbf{V}_i^{\mathbf{x}}$, where $d$ is the embedding dimension with scaling factor $\sqrt{d}$. For vector features, the dilated graph vector self-attention operates similarly but uses vector projections for queries, keys, and values: $\mathrm{DGA}_{\mathbf{x}}(i) = \mathrm{softmax}(\frac{\mathbf{A}_i^{\mathbf{x}}}{\sqrt{d}}) \mathbf{V}_i^{\mathbf{x}}$. Detailed complexity analysis is provided in Appendix E.

# 4 EXPERIMENTS

## 4.1 EXPERIMENTAL SETUP

**Dilating Distance Ranking Implementation (denoted as DKMP$^{\mathbf{R}}$).** *Dataset:* For the dilating distance ranking, we utilize the MISATO dataset (Siebenmorgen et al., 2024), containing 16,972 experimentally resolved protein-ligand complexes from the Protein Data Bank (PDB) (Berman et al., 2000). Each trajectory includes 100 snapshots spanning eight nanoseconds under constant temperature and pressure conditions. While the dataset provides time-resolved structural data and interaction energies, it omits per-snapshot potential energy and force values. Consequently, we focus exclusively on trajectory prediction with MISATO, excluding S2EF tasks. We adhere to the original dataset partitions (Siebenmorgen et al., 2024), comprising 13,765, 1,595, and 1,612 protein-ligand pairs for training, validation, and testing, respectively. Each split treats adjacent snapshots as input-output pairs, representing structures at consecutive time points $t$ and $t + 1$. To address the computational constraints in large-scale dynamic simulations, we exclude training complexes exceeding 10,000 atoms, retaining 11,807 samples for various-scale S2S evaluation. Validation and test sets remain unmodified. Besides, we include the AdK dataset to assess the performance of the dilating distance ranking strategy on the fixed-scale dataset for a comprehensive evaluation.

*Metrics*: 1) Accuracy: For the S2S task on the MISATO dataset, we apply three key metrics: Next, Final, and Average Mean Squared Error (N/F/A-MSE) (Xu et al., 2024; Wu et al., 2024). N-MSE measures the MSE between two adjacent snapshots. F-MSE is the MSE between the predicted final state and the ground truth. A-MSE, which can also be regarded as a metric for evaluating the ability to perform structure-to-trajectory (S2T) predictions, computes the MSE averaged across all discretized time steps along the decoded trajectory. 2) Distribution Similarity: We use Jensen-Shannon (JS) divergence to evaluate the distributional similarity between the predicted ensemble and the reference molecular dynamics (MD) trajectory. Specifically, JS divergence is computed in the projected feature space defined by the two slowest Time-lagged Independent Components (TIC) (Yu et al., 2025).

**Dilating Radius Cutoff Interval Implementation (denoted as DKMP$^{\mathbf{C}}$).** *Dataset*: To evaluate the proposed dilating radius cutoff interval strategy, we used Chignolin (Wang et al., 2023a), a protein consisting of 166 atoms. The dataset contains two million conformations computed at the DFT level

---

[1] For simplicity we refer to $X^t$ as $X$ in the subsequent discussion, unless stated otherwise.

Table 1: MSE (Including Next, Averaged, and Final Snapshot), and Time Consumption per Snapshot for Large-Scale Protein-Ligand Dynamics Simulations on Training Set and Testing Set of MISATO, respectively.

| Type | Method | Training Set (Avg. atoms/mol: 4880.11) † | | | | | Testing Set (Avg. atoms/mol: 4724.84) † | | | | |
|---|---|---|---|---|---|---|---|---|---|---|---|
| | | N-MSE | F-MSE | A-MSE | JS-TIC | Time (s) | N-MSE | F-MSE | A-MSE | JS-TIC | Time (s) |
| Numeric MD | AIMD | - | - | - | - | 8.64E+06‡ | - | - | - | - | 8.64E+06‡ |
| | CHARMM27 | - | - | - | - | 1.05E+04‡ | - | - | - | - | 1.05E+04 ‡ |
| ML-based MD | EGNN | 346.72 | 1874.37 | 1412.61 | 0.71 | 1.95 | 329.44 | 2933.05 | 2451.18 | 0.74 | 2.25 |
| | ViSNet | 892.12 | 1485.47 | 1034.86 | 0.69 | 2.18 | 908.06 | 1713.55 | 1153.83 | 0.75 | 3.52 |
| | LSRM | 794.12 | 1323.04 | 921.34 | 0.72 | 1.72 | 809.15 | 1525.59 | 1027.76 | 0.80 | 2.39 |
| | ESTAG | 344.62 | 363.19 | 352.11 | 0.69 | 1.09 | 308.69 | 306.40 | 310.88 | 0.72 | 1.23 |
| | FreeCG | 552.37 | 978.98 | 854.02 | 0.71 | 2.32 | 407.70 | 701.69 | 918.91 | 0.82 | 2.56 |
| | EGNO | 4.89 | 91.14 | 45.51 | 0.29 | 1.18 | 5.76 | 115.38 | 43.94 | 0.31 | 1.21 |
| | DKMP$^C$ | 7.97 | 122.82 | 67.24 | **0.38** | 1.89 | 9.536 | 135.25 | 84.25 | **0.41** | 2.01 |
| | DKMP$^R$ | **0.88** | **53.07** | **28.18** | **0.14** | 1.00 | **0.92** | **64.56** | **19.75** | **0.20** | **1.01** |

†: The dataset filtered out molecules with more than 10,000 atoms.
‡: Estimated time from (Wang et al., 2024a).

and presents notable challenges. Following (Li et al., 2024), we utilize a 9,543-structure subset in our experiments and divided the dataset into training, validation, and test sets using an 8:1:1 ratio. Chignolin is an ideal benchmark for evaluating comparatively large molecular dynamics in S2EF tasks. Additionally, we extend our evaluation to the MD22 (Chmiela et al., 2023) dataset to assess methodological generalizability. *Metrics*: For both Chignolin and MD22 datasets, we report mean absolute error (MAE) values for energy predictions and force components. Detailed descriptions of datasets and training protocols are provided in Appendix F.

**Baselines**. To comprehensively compare performance, we include a series of ML methods that can perform MD simulation, including PaiNN (Schütt et al., 2021), Equivariant Transformer (Thölke & De Fabritiis, 2022), GemNet-OC (Gasteiger et al., 2022), ClofNet (Du et al., 2022), Allegro (Musaelian et al., 2023), NequIP (Batzner et al., 2022), MACE (Batatia et al., 2022), Equiformer (Liao & Smidt, 2023), EquiformerV2 (Liao et al., 2024), ViSNet (Wang et al., 2024c), QuinNet (Wang et al., 2023b), LSRM (Li et al., 2024), EGNO (Xu et al., 2024), Ewald (Kosmala et al., 2023), and Neural P$^3$M (Wang et al., 2024b). For the task of large-scale protein-ligand dynamics simulation, we choose EGNN, VistNet, ESTAG (Wu et al., 2024), FreeCG (Shao et al., 2025), and EGNO (Xu et al., 2024) as representative methods for training. Notably, other algorithms are either incapable of running due to the out-of-memory with the same setting of computational resources (MACE, GemNet-OC, Allegro, NequIP, Equiformer (v1/v2), and FreeCG), or produce weights with NaN even with shallow layers (Equivariant Transformer).

## 4.2 RESULTS AND ANALYSIS

**Performance of DKMP$^R$.** We present performance evaluations on the filtered MISATO training and test sets in Table 1, demonstrating that our DKMP$^R$ achieves SOTA performance in next-step, final-step, and average trajectory MSE. *1)*: Our method attains atomic-scale precision (MSE < 1.0) on both training and test sets, with a six-fold error reduction compared to suboptimal baselines, confirming its ability to model dense atomic interactions. While EGNO and DKMP$^C$ surpass other baselines, their suboptimal performance reveals the difficulty in modeling various-scale molecular systems. *2)* Baseline methods exhibit catastrophic failures on the test set (e.g., final-step MSE > 100), signifying model collapse in long-term MD tasks. In contrast, our method maintains a comparatively small MSE, though the final-step MSE increases (highlighting unresolved challenges in long-temporal and large-scale MD simulations, which we reserve for future work). Besides, the results indicated by JS-TIC demonstrate that the proposed method achieves improved distributional similarity for the long-term MD simulations. *3)* The LSRM framework is a VisNet-based method using fragmentation for large-scale MD simulations. However, we identified that the BRICS fragmentation method implemented within LSRM is incapable of fragmenting certain ligands in the MISATO dataset. We then replace it with $K$-means clustering following the suggestion outlined in (Li et al., 2024). Notably, the next-step prediction errors of LSRM and its underlying backbone network (VisNet) on the protein-ligand dataset exceed those of our model by a factor of approximately 879.5 to 1012.5. These results corroborate the limitations discussed in Section 1, where existing methods are shown to struggle with capturing dense dependencies in simulating the dynamics of bio-complexes with heterogeneous molecules. *4)* Our method shows a significant improvement in efficiency compared to classical and *ab initio* MD methods while achieving the fastest inference speed among machine learning approaches.

We then evaluate our method on the testing split without any filtering. This split contains significantly larger molecular complexes, with an average of 7,151 atoms and systems exceeding 40,000 atoms. As system size increases, all baseline methods encounter substantial computational bottlenecks during structural prediction tasks and fail to complete dy-

Table 2: MSE (Including Next, Averaged, and Final Snapshot) and Time Consumption per Snapshot on Test Set of MISATO. All compared baselines failed due to out-of-memory.

| Complexes | # Atoms | Method | N-MSE | F-MSE | A-MSE | Time (s) |
|---|---|---|---|---|---|---|
| 5DU4 | 11,122 | DKMP$^R$ | 0.87 | 30.05 | 14.43 | 2.88 |
| 1YKP | 40,798 | DKMP$^R$ | 0.74 | 206.37 | 105.56 | 18.43 |
| Testing Set† | 7151.38 | DKMP$^R$ | 1.39 | 41.66 | 22.16 | 1.80 |

†: Complete testing dataset split.

namic simulations due to out-of-memory (OOM) errors. In contrast, our method maintains near-atomic accuracy across the entire testing dataset. For case studies, we analyze two representative protein-ligand complexes from MISATO: 5DU4 (11,122 atoms) and 1YKP (40,798 atoms). Notably, even for the largest system (1YKP), our method achieves atomic-level precision in next-step prediction, highlighting its capability to model large-scale interatomic interactions with full atomic resolution. Visualizations of both molecular systems are provided in Appendix G.

Furthermore, we evaluate the dilating distance ranking strategy with a fixed-scale S2S benchmark with the AdK dataset (Seyler & Beckstein, 2017), as detailed in Appendix J. Our method demonstrates consistent accuracy gains across both backbone (855 atoms) and full-atom (3,341 atoms) MD simulation configurations. These findings further confirm the effectiveness of our approach in addressing large-scale MD challenges.

Table 3: MAE of energy (kcal/mol), force (kcal/(mol·Å)), training time consumption (s), and training memory consumption (GiB) on Chignolin compared with the state-of-the-art algorithms.

| Method | Energy MAE | Force MAE | Time | GPU Memory |
|---|---|---|---|---|
| PaiNN | 0.455 | 0.605 | 0.017 | 6.206 |
| ET | 1.215 | 0.579 | 0.032 | 9.983 |
| GemNet-OC | 3.775 | 0.590 | 0.142 | 28.829 |
| ClofNet | 2.272 | 0.639 | 0.155 | 8.096 |
| Allegro | 1.660 | 0.783 | 0.186 | 34.157 |
| NequIP | 0.490 | 0.823 | 0.234 | 22.433 |
| MACE | 0.604 | 0.291 | 0.179 | 35.606 |
| Equiformer | 1.097 | 0.212 | 0.267 | 15.311 |
| EquiformerV2 | 0.898 | 0.195 | 0.279 | 23.382 |
| ViSNet | 2.436 | 0.372 | 0.046 | 11.204 |
| LSRM | 0.669 | 0.187 | 0.087 | 8.618 |
| Ewald | 0.495 | 0.294 | 0.192 | 11.946 |
| Neural P$^3$M | 0.454 | 0.261 | 0.148 | 13.195 |
| DKMP$^C$ | **0.291** | **0.126** | **0.016** | 10.835 |

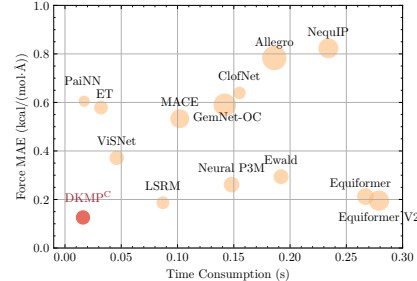

Figure 4: The comparison of model performance (y-axis), inference time consumption (x-axis), and training memory consumption (volume) among our DKMP$^C$ and other ML MD methods on Chignolin.

**Performance of DKMP$^C$.** To evaluate the effectiveness of dilated radius cutoff intervals, we conducted experiments on two standard single-scale datasets, MD22 and Chignolin. All compared methods were evaluated under the same setting, with performance and efficiency metrics summarized in Table 3. Our method achieves SOTA MAE values for both energy and force predictions in Chignolin protein dynamics simulations. Notably, the proposed approach effectively models dense interatomic interactions, yielding MAE reductions of 35.90% and 32.62% for energy and force predictions, respectively, compared to prior SOTA baselines. Furthermore, our dilated $K$-star message passing framework maintains faster inference speeds while delivering superior predictive accuracy. A comparative visualization of model performance (force MAE), inference time, and training memory consumption is provided in Figure 4. As illustrated, our method achieves an optimal trade-off between force prediction accuracy and computational efficiency, demonstrating the effectiveness of the dilated graph architecture in balancing the efficiency and accuracy for modeling comparatively large atomic systems. Besides, we evaluate our DKMP$^C$ framework on the MD22 benchmark dataset, a standard benchmark for multiscale molecular MD simulations, with systems ranging from 42 to 370 atoms. As shown in Table 9 in Appendix K, our method consistently outperforms existing approaches across all energy and force prediction tasks for comparatively large molecules (atoms > 100). Notably, our model also achieves top-tier performance on small-scale molecular systems. With the exception of force predictions for two small molecules, where it ranks second, our model attains the best performance across all tasks. Furthermore, we observed that Ewald-based methods (Ewald and Neural P$^3$M) exhibit competitive performance in modeling large atomic systems by

enhancing long-range interaction modeling. This approach is orthogonal to our method of sparsifying dense message passing; therefore, we will explore integrating these methods into our dilated message passing framework in future work.

**Molecular Dynamics Simulation on Chignolin.** To evaluate the capability of our model in MD simulations, we integrated it with the ASE (Larsen et al., 2017) toolkit and performed MD on the Chignolin peptide. First, we carried out a constant-energy (NVE) simulation at 300 K using the Velocity Verlet integrator for 100 ps with a timestep of 1 fs. Second, we report the temporal variations in total energy relative to the equilibrium value in Figure 5. The results show that our approach exhibits minimal energy drift—comparable to classical MD—over the 100-ps trajectory, indicating improved stability relative to other ML-based methods.

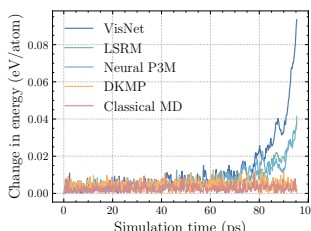

Figure 5: Energy Changes of VisNet, LSRM, Neural P$^3$M, and our DKMP$^C$ in NVE MD Simulations on Chignolin.

**Parameter Analysis.** We evaluate the impact of layer count and maximum radius cutoff value on the performance of PaiNN, VisNet, LSRM, and our proposed DKMP$^C$ architecture, respectively.

*Increasing radius cutoff.* As shown in Figure 6 (a), enlarging the radius cutoff introduces training challenges, as all baselines exhibit limited representation capacity when processing extensive neighbor lists (Li et al., 2024). While our method demonstrates less performance degradation, the universal decline across methods confirms that expanding the cutoff is incapable of learning large-scale atomic systems, validating our hypothesis in the introduction.

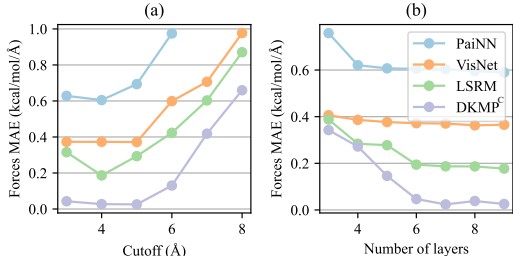

Figure 6: Comparative Studies on the Cutoff and the Number of Layers for PaiNN, VisNet, LSRM, and our DKMP$^C$ on Chignolin Dataset.

*Increasing MPNN depth.* Figure 6 (b) reveals that DKMP$^C$ successfully models dense interatomic interactions through deeper architectures. In contrast, baseline performance plateaus in sub-optimum with additional layers. Notably, the 5-layer DKMP$^C$ surpasses the 9-layer VisNet, indicating that conventional methods suffer from over-squashing and over-smoothing at greater depths. Our dilated $K$-star message passing framework circumvents these issues, enabling accuracy gains through depth scaling. In summary, this analysis demonstrates that neither radius cutoff extension nor naive layer stacking adequately models large-scale molecular interactions. The proposed dilation mechanism addresses this limitation without resorting to coarse-grained approximations, making it suitable for simulating the dynamics of large-scale molecular systems.

## 5 CONCLUSION

In this paper, we focus on large-scale molecular dynamics simulations. Motivated by the challenge of learning dense interactions for large atomic systems, we proposed a new message passing framework based on dilating star-structured message passing. Our proposed DKMP* are constructed progressively along dilated regions to capture the effects of distant atoms, resolving the issue of over-squashing. In particular, we instantiate our framework with two implementations: dilating radius cutoff intervals and distance ranking for different types of single-scale and various-scale MD simulations, respectively. Experimental results on several standard datasets (MD22, Chignolin, AdK, and MISATO) demonstrate that our method achieves state-of-the-art performance on large-scale MD tasks while maintaining optimal computational efficiency. To the best of our knowledge, this is the first method capable of simulating large-scale protein-ligand interactions with atomic-level accuracy.

ETHICS STATEMENT

The research presented in this paper adheres to the ICLR Code of Ethics. We have carefully considered the ethical implications of our work, particularly in relation to societal impacts, which are thoroughly discussed in Appendix M. Our methodology does not involve human subjects, sensitive data, or applications with a high risk of misuse. We have ensured that our models and findings are developed with fairness and transparency in mind, and we have addressed potential biases in the dataset and model design in the referenced appendix. Additionally, there are no conflicts of interest or sponsorship concerns that could compromise the integrity of this research. The use of LLMs is discussed in Appendix N.

REPRODUCIBILITY STATEMENT

To ensure the reproducibility of our results, we have provided an anonymous downloadable source code in the supplementary materials. For theoretical results, complete proofs of all claims are included in Appendix C. This paper utilizes four publicly available datasets: MISATO, AdK, Chignolin, and MD22, which can be accessed at the following locations: MISATO (https://zenodo.org/records/7711953), AdK (https://www.mdanalysis.org/MDAnalysisData/adk_equilibrium.html), Chignolin (https://doi.org/10.6084/m9.figshare.22786730), and MD22 (https://www.openqdc.io/datasets/md22). Details of data splits are provided in the Experiments section, and all relevant parameters are documented in Appendix F. These resources collectively enable full replication of our experiments and findings.

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

# TECHNICAL APPENDICES AND SUPPLEMENTARY MATERIAL

## A  RELATED WORKS

### A.1  MOLECULAR DYNAMICS SIMULATION

Equivariant graph neural networks (EGNNs) (Satorras et al., 2021; Liao et al., 2024; Du et al., 2022) represent a foundational approach for molecular dynamics simulations. Extending EGNNs, GMN (Huang et al., 2022) introduces a multi-channel framework tailored to physical dynamics by explicitly incorporating geometric constraints, such as chemical bonds. GemNet (Gasteiger et al., 2021) enhances the invariant DimeNet (Gasteiger et al., 2020) architecture by integrating dihedral

angle information. Subsequent advancements, such as PaiNN (Schütt et al., 2021) and the equivariant transformer (Thölke & De Fabritiis, 2022), employ vector embeddings to implicitly scalarize angular representations through inner products of these embeddings. Further innovations, including NequIP (Batzner et al., 2022), Allegro (Musaelian et al., 2023), and MACE (Batatia et al., 2022), leverage higher-order geometric tensors to attain superior accuracy across diverse molecular dynamics benchmarks. Equiformer (Liao & Smidt, 2023) and EquiformerV2 (Liao et al., 2024) adopt transformer-based architectures, achieving remarkable results on material science datasets. Despite these advances, existing methods predominantly target small-scale molecular systems (typically fewer than 100 atoms) and face significant challenges in scaling to larger systems.

## A.2 Large-scale Molecular Dynamics Simulation

Machine learning has shown great potential to improve the performance of large-scale MD simulations, in both accuracy and computational efficiency (Wang et al., 2024c;a). To simulate the large-scale molecular systems, several coarse-grained force field methods (Marrink et al., 2007; Kmiecik et al., 2016) have been developed, enabling the extension of time and size scales accessible to MD simulations. Such coarse modeling can improve computational efficiency at the cost of accuracy degradation (Marrink et al., 2007; Kmiecik et al., 2016). A widely adopted approach for modeling protein dynamics involves simulating only the backbone atoms, *i.e.* $C_{\alpha}$, $C$, $N$, and $O$ (Wu et al., 2024). Another simplification strategy is to exclude the hydrogen atoms from the simulations (Han et al., 2022). Recently, fragmentation-based methods, originally utilized to facilitate the quantum mechanical computations of large molecular systems, have emerged as promising solutions to address complex scaling challenges (Wang et al., 2024a; Li et al., 2024). Nevertheless, existing coarse-grained methods remain inadequate for simulating protein-ligand dynamics due to the inherent heterogeneity between proteins and ligands. In addition to these techniques, AlphaNet (Yin et al., 2025) introduces learnable local frames with spatial and temporal contractions to enhance the expressiveness of equivariant interatomic potentials; however, its evaluations are primarily limited to materials datasets, whose system sizes are significantly smaller than those of protein–ligand complexes.

Beyond the computational efficiency in modeling large-scale atomic systems, another important aspect is capturing long-range interactions. Recent works based on Ewald summation (Kosmala et al., 2023; Attali et al., 2024) provide a principled way for capturing long-range interactions, but they introduce substantial computational overhead due to the additional message passing carried out in the reciprocal space, hindering their scalability to large atomic systems. Our work is orthodox with this type of works; we would explore integrating Ewald summation to our framework for further improvement.

## A.3 Protein-Ligand Complex

Recent advancements in machine learning have profoundly influenced the study of protein-ligand interactions. For example, Equibind is a geometric deep learning model that employs equivariant neural networks to achieve accurate predictions of drug binding structures (Stärk et al., 2022). Similarly, DiffDock, a diffusion-based molecular docking method, incorporates rotational and translational transformations to enhance docking precision (Corso et al., 2023). In a related effort, E3Bind is an end-to-end equivariant network tailored for protein-ligand docking, which demonstrated state-of-the-art performance in binding affinity prediction (Zhang et al., 2023). Furthermore, TANKBind proposed by (Lu et al., 2022) is a trigonometry-aware neural network that significantly improves the prediction of drug-protein binding structures. Collectively, these studies underscore the transformative role of machine learning in elucidating and predicting protein-ligand interactions, thereby accelerating drug discovery efforts.

Despite these advancements, a critical research gap persists in modeling the dynamic aspects of protein-ligand interactions. Existing models often struggle to capture the temporal complexity and dynamic behavior of these interactions, which are essential for a comprehensive understanding of binding mechanisms and accurate affinity predictions (Siebenmorgen et al., 2024). This limitation highlights the urgent need for more advanced computational frameworks capable of simulating and analyzing the dynamic evolution of protein-ligand complexes over time.

### A.4 GNN on Learning Long-Range Dependency

Recent advances in graph neural networks (GNNs) have substantially improved the handling of long-range dependencies, a persistent challenge in graph representation learning (Li et al., 2024; Di Giovanni et al., 2023). Diverse methodologies have emerged to mitigate this limitation, encompassing weight-space regularization (Gravina et al., 2023), transformer-based architectures (Shi et al., 2021; Maskey et al., 2023), differential equation-inspired deep graph networks (Heilig et al., 2025), and graph rewiring techniques (Gutteridge et al., 2023). Within the rewiring paradigm, approaches like SDRF (Topping et al., 2022b), GRAND (Chamberlain et al., 2021a), BLEND (Chamberlain et al., 2021b), and DRew (Gutteridge et al., 2023) dynamically adjust edge connectivity during preprocessing to optimize information flow via graph densification. While our method aligns with this rewiring category, existing techniques may inadvertently introduce computational overhead during message propagation due to increased graph density (Heilig et al., 2025).

## B Notations

Table 4 summarizes the mathematical symbols and notation conventions used throughout this paper.

Table 4: List of Notations

| | |
|---|---|
| DKMP | Our framework: Dilated K-star Message Passing. |
| DKMP$^C$ | DKMP with dilated radius **c**utoff intervals. |
| DKMP$^R$ | DKMP with dilated distance **r**anking. |
| $\mathcal{G}$ | Molecular graph. |
| $G$ | Message-passing graph. |
| $N$ | Number of atoms. |
| $X$ | Atomic coordinates. |
| $H$ | Atomic features (e.g., atomic numbers). |
| $V$ | Node set. |
| $E$ | Edge set. |
| $L$ | Total number of layers. |
| $d$ | Hidden dimension. |
| $\mathcal{C}$ | Maximum cutoff for DKMP$^C$. |
| $\mathcal{M}$ | Maximum number of neighbors for DKMP$^R$. |
| $\hat{X}, S$ | Embeddings of atomic coordinates and atomic numbers. |
| $\mathbf{Q}^{s/x}, \mathbf{K}^{s/x}, \mathbf{A}^{s/x}$ | Query, key, and attention weight matrices. |
| $\mathbf{A}$ | Adjacency matrix. |
| $\mathcal{N}_K^{(l)}(i)$ | Neighborhood of atom $i$ in the $l$-th MPNN layer. |
| $\mathcal{N}_C^{(l)}(i)$ | Neighborhood of atom $i$ under the dilated cutoff interval. |
| $\mathcal{N}_R^{(l)}(i)$ | Neighborhood of atom $i$ under dilated distance ranking. |

## C DKMP* in Long-Range Modeling

The hidden feature $\mathbf{z}_i^{(l)} = h_i^{(l)}(\mathbf{v}_1, \ldots, \mathbf{v}_n)$, computed by an MPNN with $l$ layers, constitutes a differentiable function of the input node features $\{\mathbf{v}_1, \ldots, \mathbf{v}_n\}$ provided the update and message functions $\phi_l$ and $\psi_l$ are differentiable. Over-squashing arises when a node representation $\mathbf{z}_i^{(l)}$ fails to incorporate information from an input feature $\mathbf{v}_s$ of a node $s$ located at distance $r$ from node $i$ (Topping et al., 2022a). This phenomenon can be quantified as:

$$\left| \frac{\partial \mathbf{z}_i^{(r+1)}}{\partial \mathbf{x}_s} \right| \leq (\alpha\beta)^{r+1} (\mathbf{A}^{r+1})_{is}, \tag{5}$$

where $i, s \in V$ with $s \in \mathcal{N}_{r+1}(i)$, and the constants satisfy $|\nabla\phi_l| \leq \alpha$ and $|\nabla\psi_l| \leq \beta$ for $0 \leq l \leq r$. Critically, over-squashing manifests when $\mathbf{A}^{r+1}$ induces an exponentially decaying dependence

of $\mathbf{z}_i^{(r+1)}$ on features at distance $r$. In contrast, our proposed DKMP* framework enables direct message passing between nodes separated by $r$ hops, bypassing the need for iterative powers of $\mathbf{A}$ and mitigating exponential decay.

# D MODEL DETAILS FOR DILATING CUTOFF INTERVAL

## D.1 INPUT LAYER

Given atomic coordinates and type features $X^t = [\mathbf{x}_1; \cdots; \mathbf{x}_N]$ and $H = [\mathbf{h}_1; \cdots; \mathbf{h}_N]$, where $\mathbf{x}_i \in \mathbb{R}^3$ denotes Cartesian coordinates and $\mathbf{h}_i$ represents one-hot encoded atomic types, we first project the atomic types into a latent space via an embedding layer:

$$\mathbf{z}_i = \text{Embedding}(\mathbf{h}_i) \in \mathbb{R}^d,$$

where $d$ is the latent space dimension. For neighbors $\mathcal{N}_l^C$ of atom $i$, identified by Eq. (3) in the $l$-th message-passing layer, we compute displacement vectors $\mathbf{x}_{ij} = \mathbf{x}_j - \mathbf{x}_i$. These are converted to irreducible representations by applying real spherical harmonics to their unit vectors:

$$\mathbf{r}_{ij} = Y^{(1)}\left(\frac{\mathbf{x}_{ij}}{\|\mathbf{x}_{ij}\|}\right) \oplus Y^{(2)}\left(\frac{\mathbf{x}_{ij}}{\|\mathbf{x}_{ij}\|}\right) \in \mathbb{R}^{3+5},$$

where $Y^{(l)}$ denotes the $l$-th order real spherical harmonics. The Euclidean norms $\|\mathbf{x}_{ij}\|$ are mapped to high-dimensional scalar edge features via radial basis functions:

$$\mathbf{f}_{ij} = \text{RBF}(\|\mathbf{x}_{ij}\|) \in \mathbb{R}^d.$$

## D.2 MESSAGE PASSING FOR DILATING CUTOFF INTERVAL

For the first message passing layer, the $\mathbf{v}_i^l$ is the sum of all unit vectors from node $i$ to its all neighboring nodes $j$, where node $i$ is the intersection of all unit vectors (Wang et al., 2024c). For each layer, $\mathbf{r}_{ij}$ is the irreducible representation for edge from node $i$ to its neighboring node $j$. In summary, the key operations in our dilated message passing are given as follows:

$$m_i^l = \sum_{j \in \mathcal{N}_c(i)} \phi_s^l(\mathbf{z}_i, \mathbf{z}_j, \mathbf{f}_{ij}), \tag{6}$$

$$\mathbf{m}_i^l = \sum_{j \in \mathcal{N}_c(i)} \phi_v^l(m_{ij}^l, \mathbf{r}_{ij}, \mathbf{v}_j), \tag{7}$$

$$\mathbf{v}_i^{l+1} = \phi_u^l(\mathbf{v}_i^l, m_i^l, \mathbf{m}_i^l). \tag{8}$$

We provide model details for dilating radius cutoff interval in Appendix D.

The primary distinction between dilated message passing method and traditional graph message passing is that we forgo updating edge information. Instead, both scalar and vector edge features are recalculated in each layer of the network based on dynamically modeled neighbors. This dynamics and direct learning of long-distance interatomic interactions avoids the issues of long-range dependencies, aligning more closely with physical principles. We present an illustration for dilating cutoff interval in Fig. 7.

## D.3 OUTPUT LAYERS

DKMP* with dilated cutoff intervals are designed for **S2EF** task and is therefore designed based on energy-conservative field, which means we derive the force from the predicted potential energy. The total energy $e$ of the molecule is the sum of the last-layer node features via an equivariant gated module (Wang et al., 2024c; Shao et al., 2025). And the force is the negative gradients of the total energy:

$$F_i = -\nabla_{\mathbf{x}_i} e. \tag{9}$$

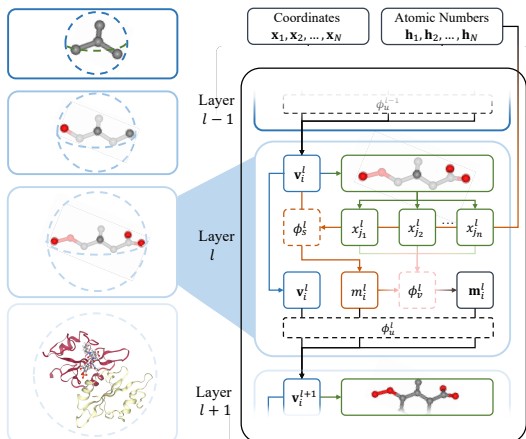

Figure 7: Schematic Representation of the DKMP[C] Framework: Implementing Dilating Cutoff Intervals as a Specific Instantiation

## E  TIME COMPLEXITY ANALYSIS

The DKMP* algorithm relies on cutoff-based interval and distance ranking computations, which can be precomputed with a worst-case complexity of $\mathcal{O}(N + KL \log N)$. Once precomputed, these values are reusable across all simulation runs. The dilated distance ranking mechanism, designed to model protein-ligand interaction dynamics, operates under the condition $KL \ll |V|$, ensuring that local $KL$-nearest neighbor relationships exhibit negligible variation during simulations. Consequently, we compute the distance ranking order for each atom only for the initial structural configuration and reuse it across all subsequent trajectory frames.

For dilated graph attention, the QKV linear projections require $3Nd$ FLOPs. In dilated attention, $\mathbf{A}_i$ has dimensions $K$, and computing it for all atoms incurs a cost of $\mathcal{O}(NKd)$. The tensor $\mathbf{V}_i^{\mathbf{x}}$, with dimensions $N \times K \times d$, also requires $\mathcal{O}(NKd)$ operations to apply the attention weights. In contrast, traditional graph attention with full connectivity scales as $\mathcal{O}(N^2 d)$. Since $K \ll N$, our dilated attention mechanism significantly reduces the computational burden in large-scale MD simulations. For the specific implementation, we refer to the C++ and CUDA kernels introduced in neighborhood attention transformer (Hassani et al., 2023) to further accelerate the model architecture.

## F  TRAINING DETAILS

The training details are outlined below, with dataset-specific parameters provided in Table 5.

**Training**

1. Optimizer: Adam (Kingma & Ba, 2015) optimizer is used with a constant learning rate of $10^{-4}$ as our default training configuration.

2. GPU: NVIDIA GeForce RTX 3090

3. CPU: Intel(R) Xeon(R) Platinum 8338C CPU

4. Memory: 512 GB

## G  VISUALIZATION OF EXPERIMENTAL RESULTS ON MISATO DATASET

We present a structural visualization of the protein-ligand complexes **5DU4** and **1YKP** in Figure 8. Our experimental results demonstrate that the proposed method effectively models large-scale biomolecular complexes while achieving highly accurate predictions of their structural conformations at subsequent timesteps. This capability underscores the framework's potential for advancing learning-based large-scale molecular dynamics simulations.

Table 5: Dataset Information and Training Details

| Task | S2EF | | S2S | | |
|---|---|---|---|---|---|
| Dataset | MD22 | Chignolin | AdK-Backbone | AdK-Full-Atom | MISATO |
| Atoms | 42-370 † | 166 | 855 | 3341 | 556-40,798 |
| Batch size | 4-8 | 4 | 32 | 8 | 2 |
| Epochs | 2,000 | 2,000 | 1500 | 1500 | 10 |
| Time (h) | 10 | 24 | 24 | 48 | 120 |

†: Following standard protocol (Chmiela et al., 2023; Wang et al., 2024c), individual models were trained for each molecular system on the MD22 benchmark.

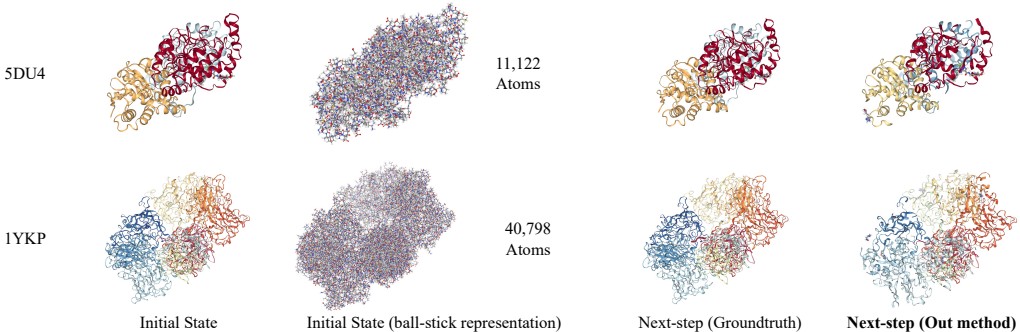

Figure 8: Visualization of Simulation Results on Protein-Ligand **5DU4** and **1YKP**.

## H EXPERIMENT ON EQUIVARIANCE OF DKMP$^R$

To assess the equivariance properties of the DKMP$^R$ model, we conducted experiments on the MIS-ATO dataset by applying random rotations to the test set and evaluating performance using the same DKMP$^R$ model. Although the rotated datasets exhibit a slight performance degradation, the model remains competitive relative to existing baselines. Moreover, given the substantial computational overhead typically associated with incorporating strict equivariance into model architectures, this level of performance represents a practical and efficient trade-off.

Table 6: Performance of DKMP$^R$ on the original and randomly rotated MISATO test sets, evaluating the model's robustness to rotational perturbations.

| Splits | Selected Testing Set (Avg. atoms/mol: 4724.84) | | | | | Complete Testing Set | | |
|---|---|---|---|---|---|---|---|---|
| Metrics | N-MSE | F-MSE | A-MSE | JS-TIC | Time (s) | N-MSE | F-MSE | A-MSE |
| MISATO | **0.92** | **64.56** | **19.75** | 0.20 | 1.01 | **1.39** | **41.66** | **22.16** |
| MISATO (rotated) | 0.94 | 65.46 | 20.18 | 0.20 | 1.01 | 1.42 | 42.16 | 22.75 |

## I EXPERIMENTS ON ABLATION STUDY

In this section, we conducted three ablation experiments in Table 7 to provide a comprehensive understanding of the design of our DKMP*:

1. Using a single KNN graph with $K = \mathcal{M}$ (i.e., identical total edge count as L dilated layers) and $L$ layers yields poor performance and out-of-memory in the full atom AdK dataset. This highlights the benefit of the physically motivated dilation strategy, which distributes interactions across layers and facilitates more efficient message passing.

2. Random partitioning leads to a clear degradation in accuracy, confirming that distance-based hierarchical grouping is essential for maintaining geometric coherence and effective message passing.

Table 7: Ablation Study on DKMP$^R$

| Variants | Backbone | Full atoms |
|----------|----------|------------|
| 1. DKMP$^R$ (one-shot KNN) | 2.739 | OOM |
| 2. DKMP$^R$ (random partition) | 2.962 | 3.253 |
| 3. DKMP$^R$ (allow 25 % edge reuse) | 2.383 | 2.740 |
| DKMP$^R$ | **1.900** | **2.715** |

3. Allowing 25% edge reuse produces reasonable but inferior results. Since the model depth remains unchanged, we attribute this performance drop to reduced overall geometric coverage when edges repeat across layers instead of expanding the receptive field.

## J    EXPERIMENT ON S2S TASK WITH ADK EQUILIBRIUM TRAJECTORY DATASET

To rigorously evaluate our methodology, we further employ the adenylate kinase (AdK) equilibrium trajectory dataset integrated within the MDAnalysis toolkit (Gowers et al., 2019), which captures the molecular dynamics of apo adenylate kinase. In detail, the AdK equilibrium dataset depicts the molecular dynamics trajectory of apo adenylate kinase with CHARMM27 force field (MacKerell Jr et al., 2000), simulated with explicit water and ions in NPT at 300 K and 1 bar. The meta-data is saved every 240 ps for a total of 1.004. Our evaluation employs both backbone and full-atom configurations while adhering to established data splits from prior studies (Han et al., 2022; Xu et al., 2024), which divides the entire trajectory into a training set with 2481 sub-trajectories, a validation set with 827, and a testing set with 878 trajectories, respectively.

The experimental results, summarized in Table 8, demonstrate that our DKMP$^R$, enhanced by dilated message passing, achieves SOTA performance on this challenging benchmark. Notably, DKMP$^R$ delivers statistically significant improvements in accuracy for both backbone and full-atom modeling tasks. These results validate the efficacy of dilated $K$-star message passing in capturing long-range interatomic interactions and underscore its broader applicability to large-scale molecular dynamics simulation scenarios.

Table 8: F-MSE on AdK equilibrium trajectory dataset.

| AdK | Linear | RF | MPNN | EGNN | EGHN | EGNO | DKMP$^R$ |
|-----|--------|-----|------|------|------|------|----------|
| Backbone | 2.8900 | 2.8460 | 2.3220 | 2.7350 | 2.234 | 2.231 | **1.900** |
| Full-atom | - | - | - | - | 2.882 | 2.866 | **2.715** |

"**-**" indicates that the model runs out of memory on full atomic modeling.

## K    EXPERIMENT ON S2EF TASK WITH MD22 DATASET

We evaluate our DKMP$^C$ framework on the MD22 benchmark dataset and present the results in Table 9. The consistent achievement of the lowest MAE in both energy and force predictions for most molecules underscores the robustness and generalizability of DKMP$^C$. These findings highlight the critical role of sparsifying modeling interactions in machine learning approaches, enabling efficient and accurate simulations for large-scale molecular dynamics applications. Furthermore, the results suggest that DKMP$^C$ effectively captures complex molecular interactions even in large-scale atomic systems.

Table 9: Performances on MD22 dataset. The results are reported in MAE. The energies and forces are measured in kcal/mol and kcal/(mol·Å), respectively. The best numbers are marked in **bold**.

| Molecule | Atoms | | | PaiNN | MACE | Equiformer | ViSNet | LSRM | QuinNet | FreeCG | Neural P$^3$M | DKMP$^C$ |
|---|---|---|---|---|---|---|---|---|---|---|---|---|
| Ac-Ala3-NHMe | 42 | Energy | | 0.1168 | 0.0622 | 0.0828 | 0.0796 | 0.0789 | 0.0840 | 0.0507 | 0.0719 | **0.0434** |
| | | Forces | | 0.2302 | 0.0876 | 0.0804 | 0.0972 | 0.0887 | 0.0681 | **0.0531** | 0.0788 | 0.0734 |
| DHA | 56 | Energy | | 0.1151 | 0.1317 | 0.1788 | 0.1526 | 0.0862 | 0.1181 | 0.0761 | **0.0712** | 0.0761 |
| | | Forces | | 0.1355 | 0.0646 | 0.0506 | 0.0668 | 0.0534 | 0.0515 | 0.0507 | 0.0679 | **0.0493** |
| Stachyose | 87 | Energy | | 0.1517 | 0.1244 | 0.1404 | 0.1283 | 0.1252 | 0.2260 | 0.1830 | 0.0856 | **0.0852** |
| | | Forces | | 0.2329 | 0.0876 | 0.0635 | 0.0869 | 0.0632 | **0.0543** | 0.6120 | 0.0940 | 0.0767 |
| AT-AT | 60 | Energy | | 0.1673 | 0.1093 | 0.1309 | 0.1688 | 0.1007 | 0.1440 | 0.0665 | 0.0714 | **0.0639** |
| | | Forces | | 0.2384 | 0.0992 | 0.0960 | 0.1070 | 0.0881 | 0.0687 | 0.0634 | 0.0740 | **0.0619** |
| AT-AT-CG-CG | 118 | Energy | | 0.2638 | 0.1578 | 0.1510 | 0.1995 | 0.1335 | 0.3790 | 0.2540 | 0.1124 | **0.1121** |
| | | Forces | | 0.3696 | 0.1153 | 0.1252 | 0.1563 | 0.1065 | 0.1273 | 0.1252 | 0.0993 | **0.0959** |
| Buckyball catcher | 148 | Energy | | 1.1712 | 0.4989 | 0.4093 | 0.5089 | 0.3319 | 0.5630 | 0.5120 | 0.3543 | **0.3229** |
| | | Forces | | 0.6809 | 0.0853 | 0.1782 | 0.1849 | 0.1026 | 0.1091 | 0.1783 | 0.0846 | **0.0789** |
| Double-walled nanotube | 370 | Energy | | 3.5324 | 1.6782 | 0.7024 | 0.8004 | 1.8331 | 1.8100 | 0.5430 | 0.7751 | **0.4992** |
| | | Forces | | 0.5205 | 0.2767 | 0.2583 | 0.3624 | 0.3391 | 0.2473 | 0.2449 | 0.2561 | **0.2394** |

## L    LIMITATIONS

Our experiments on the MISATO dataset demonstrate that all evaluated methods, including our proposed approach, exhibit suboptimal performance on the F-MSE metric, highlighting the challenges in achieving stable MD simulations over long timescales. We defer this challenge to future research.

Since each layer processes a disjoint and fixed-size neighborhood, inter-device communication is minimal, and memory loads are distributed evenly. This architecture provides a clear path toward scaling DKMP$^R$ well beyond the regimes demonstrated so far. Future work will explore larger and more complex systems, investigate dynamic neighborhood partitioning for adaptive workloads, and benchmark performance across heterogeneous computing environments to fully realize the method's scalability potential.

## M    IMPACT STATEMENTS

This paper presents work whose goal is to advance the field of Artificial Intelligence (AI) for scientific fields, such as material science, chemistry, and biology. The obtained experience/knowledge will greatly boost AI technologies in facilitating the process of scientific knowledge discovery.

Machine learning for force fields opens up possibilities for understanding molecules at a fast speed. The potential for misuse and unintended consequences necessitates strict ethical guidelines, robust regulation, and responsible use of these technologies to prevent harm to individuals and society.

## N    THE USE OF LARGE LANGUAGE MODELS

The core method development and research ideation in this paper were conducted independently of LLMs, and LLMs did not contribute to any original or non-standard components of the work. The authors utilized LLMs solely as a general-purpose assist tool for checking grammar and improving the clarity of the manuscript, as well as for aiding in the comprehension of existing literature. All content in this submission, including any text refined with LLM assistance, has been thoroughly reviewed by the authors, who take full responsibility for its accuracy, integrity, and compliance with ethical standards. No LLMs are considered contributors or eligible for authorship.

