# OpenReview forum: "Large-Scale Molecular Dynamics Simulation: Direct Interatomic Modeling with Dilated Message Passing"
_ICLR.cc/2026/Conference — Submitted to ICLR 2026_

### Official Review · Reviewer_KRk4 · 2025-10-28

**Soundness:** 1
**Presentation:** 1
**Contribution:** 1
**Rating:** 0
**Confidence:** 5

**Summary:**

The study shows a dilated operation in message passing and benchmarked it on several datasets for energy and force predictions. Overall speaking, the design lacks novelty; the experiments cannot fully support the conclusion; some results are suspicious; many professional terms are mistanken used; and the paper is poor written. It's far from the standard of ILCR.

**Strengths:**

1. reasonable overiview and introduction on the background and latest progress in this field
2. detailed hyperparametes are shown in supplementary materials.

**Weaknesses:**

1. Lack of novelty
The dilated operation presented in this study is too simple and lacks novelty. Almost operations and equations in the model design are from ViSNet paper. The dilation on radius is just a tiny trick insteaf of a novel design as the dilation employed in CNN or other models has been demonstrated more than 10 years before.

2. Suspicous results
Table 1. There is no evaluation on MISATO dataset in the paper (Wang et al., 2024a). Where the results in "AIMD" and "CHARMM27" rows come from?
Table 2. The authors claimed all the other models suffered from OOM issue. As PaiNN and EGNN are smaller than DKMP, why they all failed?
Table 1 and Table 3. Why the compared models are inconsistent?
Table 1 and Table 3. Why neglect the SoTA models, e.g., MACE-OFF, SO3Krate, Equiformer v2 for comparison?
Table 3 and Figure 4. Why choose Equiformer rather than Equiformer v2 for comparison? Even for v2, it has been published for several years.
Figure 5. It seems that all the other models except DKMP show abnormal energy flunctuations in NVE simulations. First, it does NOT show DKMP's supriority. Instead, it means the MD settings for other models are probably WRONG! Second, even DKMP's simulation show stable energy values. It has nothing to do with the statement "throughout long-term MD simulations, more accurately capturing
the physics of real MD simulations for large molecular systems"! 100ps simulation is too short. Where's the "accurate" come from? where the "physics" come from? What does "real MD" mean? Chignolin has only 166 atoms. It has nothing to do with "large molecular system"!
The author lack basic domain knowledge on MD simulations and the experimental results are highly suspicous!

3. Cofunsed terminology
Painn, MACE, Allegro, etc...  are NOT MD simulation methods. Instead, they are machine learning force fields.
The unit of force is kcal/ (mol*Angstrom) instead of kcal/mol/Angstrom.
As far as I know, Chignolin dataset has 2 million samples, insteaf of 10 thousand. https://figshare.com/articles/dataset/_strong_AIMD-Chig_exploring_the_conformational_space_of_166-atom_protein_strong_em_strong_Chignolin_strong_em_strong_with_strong_em_strong_ab_initio_strong_em_strong_molecular_dynamics_strong_/22786730
The confused and misused terminology show the authors lack adequate knowledge in this field.

4. Poor written
The demonstration on model design is not clear. Almost operations in the model are directly from ViSNet. Too many confused terminology. And more grammer errors..

**Questions:**

The authors should seriously address the concerns shown in Weakness point by point to improve the quality of the paper.

---

> ### Author Response · Authors · 2025-11-21
> **First Response to Reviewer KRk4 [1/2]**
>
> # Response to Reviewer KRk4
> We thank the time and effort of the reviewer.
> # Response to Weakness 1
> > Lack of novelty The dilated operation presented in this study is too simple and lacks novelty. Almost operations and equations in the model design are from ViSNet paper. The dilation on radius is just a tiny trick insteaf of a novel design as the dilation employed in CNN or other models has been demonstrated more than 10 years before.
>
> Indeed, VisNet, LSRM, Equiformer, Ewald, including ours, all share the model design of equivariant graph neural networks. We want to highlight that our main contributions lie in the physically grounded multi-layer dilation mechanism to efficiently model large-scale atomic systems, beyond the pure message equations.
>
> In here, we further highlight our contributions below: Our DKMP* generalizes dilation to 3D molecular graphs, where dilation corresponds to physics-inspired interatomic distance scaling with geometric ordering and dynamic partitioning of neighbors at each layer. We also theoretically show that our method alleviates the over-squashing, a unique issue in graph learning. We also instantiated our idea into two implementations and verified its effectiveness.
>
> # Response to Weakness 2
> > Suspicous results Table 1.
>
> We have provided the code in the supplementary material; all reported results are reproducible.
>
> ## Response to Weakness 2.1
> > There is no evaluation on MISATO dataset in the paper (Wang et al., 2024a). Where the results in "AIMD" and "CHARMM27" rows come from?
>
> We follow the evaluation of the paper (Wang et al., 2024a). Specifically, we use “AIMD” and “CHARMM27” rows in Table 1 to report the estimated simulation time, instead of accuracy metrics.
>
> ## Response to Weakness 2.2
> > Table 2. The authors claimed all the other models suffered from OOM issue. As PaiNN and EGNN are smaller than DKMP, why they all failed?
>
> We would like to highlight that: The molecules in Table 2 (the complete MISATO testing set) are orders of magnitude larger than Chignolin in Table 3, which shows that PaiNN and EGNN are smaller than DKMP$^{C}$. Only DKMP$^R$, with fixed-size dilated neighbor sets, can operate under such extreme neighborhood sizes.
>
> ## Response to Weakness 2.3
> > Table 1 and Table 3. Why the compared models are inconsistent?
>
> We respectfully invite the reviewer to refer to section 4.1. The baselines for two tasks have been discussed:
> - For the task of large-scale protein-ligand dynamics simulation, we choose EGNN, VistNet, ESTAG (Wu et al., 2024), FreeCG (Shao et al., 2025), and EGNO (Xu et al., 2024) as representative methods for training. Notably, other algorithms are either incapable of running due to out-of-memory issues with the same setting of computational resources (MACE, GemNet-OC, Allegro, NequIP, Equiformer, and FreeCG), or produce weights with NaN even with shallow layers (Equivariant Transformer).
>
> Therefore, the sets of baselines differ because of the tasks and feasible models.
>
> ## Response to Weakness 2.4
> > Table 1 and Table 3. Why neglect the SoTA models, e.g., MACE-OFF, SO3Krate, Equiformer v2 for comparison?
>
> We acknowledge the existence of numerous models. Our baseline selection focuses primarily on recent methods, including VisNet (2024), LSRM (2024), ESTAG (2024), and NeuralP3M (2024), all published after the cited works (MACE-OFF 2023, SO3Krate 2022, Equiformer v2 2023).
>
> Due to substantial computational cost and model diversity, it is unfortunately infeasible to benchmark all existing SOTA models exhaustively. Nevertheless, we appreciate the suggestion and will expand the discussion in the related work section.
>
> ## Response to Weakness 2.5
> > Table 3 and Figure 4. Why choose Equiformer rather than Equiformer v2 for comparison? Even for v2, it has been published for several years.
>
> Again, we hope the reviewer could understand the difficulty of exhausting all baselines for comparison. Still, in response to the reviewer’s concern, we have added **Equiformer v2** to Table 3 and Figure 4 as an additional baseline in the revision for completeness.
>
> ## Response to Weakness 2.6
>
> > Figure 5. It seems that all the other models except DKMP show abnormal energy flunctuations in NVE simulations. First, it does NOT show DKMP's supriority. Instead, it means the MD settings for other models are probably WRONG!
>
> 1. We confirm that all MD configurations were validated and shared identical settings across models (integrator, timestep, thermostat off, initialization). The differences arise from each MLFF’s intrinsic stability, not implementation errors.
> 2. We appreciate the reviewer’s suggestion and have added a classical MD baseline (using the same integrator and settings, with the MLFF replaced by a classical force field) for comparison. The results, now included in Figure 5, show that the classical MD simulation exhibits an energy fluctuation pattern consistent with that of our method. We hope this additional evidence helps address the reviewer’s concern.

---

> ### Author Response · Authors · 2025-11-21
> **First Response to Reviewer KRk4 [2/2]**
>
> ## Response to Weakness 2.7
> > Second, even DKMP's simulation show stable energy values. It has nothing to do with the statement "throughout long-term MD simulations, more accurately capturing the physics of real MD simulations for large molecular systems"! 100ps simulation is too short. Where's the "accurate" come from? where the "physics" come from? What does "real MD" mean? Chignolin has only 166 atoms. It has nothing to do with "large molecular system"! The author lack basic domain knowledge on MD simulations and the experimental results are highly suspicous!
>
> 1. We acknowledge that the definition of 'long-term' may vary in different papers and domains. However, 100 ps are well-accepted and long enough to test the performance of ML methods for molecular dynamics simulation [1, 2]. We would revise this expression for clarity.
> 2. We understand the usage of 'accurate', 'physics', and 'real MD' may require further consideration. Our intention was simply to highlight DKMP’s improved energy stability relative to MLFF baselines, not to claim full physical fidelity.
> 3. Again, we acknowledge that the definition of 'large-scale' may vary in different papers. However, systems with >100 atoms are considered large-scale in many MLFF/MLMD papers [3,4,5]. Nonetheless, we have revised the expression for the Chignolin system for rigor.
>
>
> [1] Amin, Ishan, Sanjeev Raja, and Aditi S. Krishnapriyan. "Towards Fast, Specialized Machine Learning Force Fields: Distilling Foundation Models via Energy Hessians." The Thirteenth International Conference on Learning Representations.
>
> [2] Fu, Xiang, et al. "Forces are not Enough: Benchmark and Critical Evaluation for Machine Learning Force Fields with Molecular Simulations." Transactions on Machine Learning Research.
>
> [3] Wang, Yusong, et al. "Enhancing geometric representations for molecules with equivariant vector-scalar interactive message passing." Nature Communications 15.1 (2024): 313.
>
> [4] Li, Yunyang, et al. "Long-Short-Range Message-Passing: A Physics-Informed Framework to Capture Non-Local Interaction for Scalable Molecular Dynamics Simulation." The Twelfth International Conference on Learning Representations.
>
> [5] Kosmala, Arthur, et al. "Ewald-based long-range message passing for molecular graphs." International Conference on Machine Learning. PMLR, 2023.
>
>
> # Response to Weakness 3
> > Cofunsed terminology Painn, MACE, Allegro, etc... are NOT MD simulation methods. Instead, they are machine learning force fields. The unit of force is kcal/ (mol*Angstrom) instead of kcal/mol/Angstrom. As far as I know, Chignolin dataset has 2 million samples, insteaf of 10 thousand. https://figshare.com/articles/dataset/_strong_AIMD-Chig_exploring_the_conformational_space_of_166-atom_protein_strong_em_strong_Chignolin_strong_em_strong_with_strong_em_strong_ab_initio_strong_em_strong_molecular_dynamics_strong_/22786730 The confused and misused terminology shows the authors lack adequate knowledge in this field.
>
> 1. We agree that PaiNN, MACE, and Allegro are machine learning force fields, but they can be integrated into the numerical MD process to perform MD simulations. We will rephrase these terms throughout the paper to avoid confusion.
> 2. We fully acknowledge that the conventional unit of force is kcal/(mol·Å). While the shorthand “kcal/mol/Å” is also commonly used in many MLFF literature [1,2,3], we will adopt the precise notation in the revision.
> 3. Regarding AIMD-Chig: we follow prior work [2]in using a curated 9,543-sample subset for fair comparison. We have explicitly clarified this in the revised manuscript.
>
> [1] Wang, Yusong, et al. "Enhancing geometric representations for molecules with equivariant vector-scalar interactive message passing." Nature Communications 15.1 (2024): 313.
>
> [2] Li, Yunyang, et al. "Long-Short-Range Message-Passing: A Physics-Informed Framework to Capture Non-Local Interaction for Scalable Molecular Dynamics Simulation." The Twelfth International Conference on Learning Representations.
>
> [3] Shao, Shihao, et al. "FreeCG: Free the Design Space of Clebsch-Gordan Transform for Machine Learning Force Fields." The Thirteenth International Conference on Learning Representations.
>
> # Response to Weakness 4
> > Poor written The demonstration on model design is not clear. Almost operations in the model are directly from ViSNet. Too many confused terminology. And more grammer errors..
>
> We thank the reviewer for this feedback.
> Our intention was to present DKMP as clearly and transparently as possible. **All components** inherited from ViSNet were placed in the supplementary material deliberately—for completeness and to avoid interrupting the main narrative with well-established operations. We also ensured that terminology followed conventions used in prior literature to maintain consistency.
> That said, we acknowledge that clarity can always be improved. In the revision, we will carefully proofread the manuscript.

---

### Official Review · Reviewer_1rjP · 2025-10-30

**Soundness:** 3
**Presentation:** 4
**Contribution:** 3
**Rating:** 8
**Confidence:** 4

**Summary:**

The paper proposes Dilated K-Star Message Passing (DKMP*) for large-scale MD at full-atom resolution. The key idea is to partition edges by distance into mutually exclusive, increasingly “dilated” K-star neighborhoods and stack one-layer MPNNs over these partitions. Two concrete variants are presented: (1) DKMPC with dilated radius-cutoff intervals for single-scale systems (MD22, Chignolin); and (2) DKMPR with dilated distance-ranking + a light graph-attention core (dropping strict equivariance) for various-scale protein–ligand systems (MISATO, AdK). Results claim SOTA accuracy/efficiency, including atomic-level next-step precision on protein–ligand systems up to ~40k atoms, and improved NVE stability on Chignolin.

**Strengths:**

1. Broad evaluation: S2EF on Chignolin/MD22 and S2S on MISATO/AdK; useful mix of small and very large systems
2. Scalability: MISATO results include cases with 11k–40k atoms; baselines reportedly OOM while DKMPR runs in ≤~18 s/snapshot for the largest complex

**Weaknesses:**

1. Proposition 3.1 claims that DKMP* interactions are “immune to over-squashing.” While rewiring can mitigate message-passing bottlenecks, describing the model as “immune” is overstated unless bounded influence distortion is formally demonstrated—e.g., via curvature or flow-based analyses rather than adjacency-power arguments. I recommend tempering this claim or expanding the proof to include a modern formalism such as influence decay bounds, effective resistance, or discrete Ricci curvature, to substantiate robustness beyond intuitive edge-connectivity reasoning.
2. Additional ablations that could strengthen the empirical analysis include: (1) Edge-budget control: compare one-shot KNN with K = M versus L dilated K-stars summing to M under identical total edge counts; (2) Partition strategy: random versus distance-sorted partitioning to demonstrate the impact of ordering; (3) Hyperparameter sensitivity: vary K and L to map accuracy–latency trade-offs and reveal potential non-trivial optima; (4) Mutual-exclusion analysis: with versus without edge reuse across layers to verify its necessity; and (5) Equivariance ablation: equivariant versus non-equivariant variants on MISATO with efficiency–accuracy curve
3. The paper occasionally uses inconsistent notation (e.g., $N_{K}^{l}(i)$ vs. $N_{C}^{l}(i)$), which can obscure the hierarchy and semantics of neighborhood definitions. A unified notation scheme, accompanied by a concise schematic diagram (with notation and equations) illustrating the structure and information flow of DKMP layers, would substantially improve clarity and readability.
4. While energy conservation is qualitatively demonstrated (Fig. 5), the paper lacks quantitative physical validations such as energy drift rates or RMSD stability analyses. Including velocity autocorrelation function (VACF) plots would also provide valuable insight into the system’s dynamical fidelity.

**Questions:**

1. What is the time complexity as a function of (N, K, L, M) for both variants and contrast to dense-cutoff MPNN?
2. How does your framework handle periodic boundary conditions (PBC)? If applicable, could you demonstrate its validity on at least one standard PBC benchmark?
3. A common and informative analysis is to examine potential decay—could you plot the learned potential to verify that it exhibits the correct decaying behavior with increasing interatomic distance?
4. How does this sampling strategy ensure smooth transitions of energy and forces between consecutive MD frames? In other words, how does it avoid discontinuities caused by differences where one neighbor jumps from one edge set to a different edge set?
5. Could you clarify how the model scales to extremely large systems (e.g., tens of thousands of atoms)? The main text currently lacks sufficient technical details on the specific design or computational strategies—such as memory partitioning, neighbor sampling, or distributed message-passing—that enable efficient handling of such large systems.

---

> ### Author Response · Authors · 2025-11-21
> **First Response to Reviewer 1rjP [1/3]**
>
> # Response to Reviewer 1rjP
> We genuinely appreciate the reviewer’s thoughtful and detailed comments. We are delighted to receive the reviewer’s recognition of our work.
> # Response to Weakness 1
> >Proposition 3.1 claims that DKMP* interactions are “immune to over-squashing.” While rewiring can mitigate message-passing bottlenecks, describing the model as “immune” is overstated unless bounded influence distortion is formally demonstrated—e.g., via curvature or flow-based analyses rather than adjacency-power arguments. I recommend ...
>
> We appreciate this important observation and agree that the term “immune” is a strong word. We have revised our wording to “alleviate” rather than “immune.”
>
> # Response to Weakness 2
> > Additional ablations that could strengthen the empirical analysis include: (1) Edge-budget control: compare one-shot KNN with K = M versus L dilated K-stars summing to M under identical total edge counts; (2) Partition strategy: random versus distance-sorted partitioning to demonstrate the impact of ordering; (3) Hyperparameter sensitivity: vary K and L to map accuracy–latency trade-offs and reveal potential non-trivial optima; (4) Mutual-exclusion analysis: with versus without edge reuse across layers to verify its necessity; and (5) Equivariance ablation: equivariant versus non-equivariant variants on MISATO with efficiency–accuracy curve
>
> For (1) edge-budget control, (2) partition strategy, and (4) mutual-exclusion, we conducted targeted ablation experiments to isolate their effects:
> - Using a single KNN graph with $K=\mathcal{M}$ (i.e., identical total edge count as L dilated layers) and $L$ layers yields poor performance and out-of-memory in the full atom AdK dataset. This highlights the benefit of the physically motivated dilation strategy, which distributes interactions across layers and facilitates more efficient message passing.
> - Random partitioning leads to a clear degradation in accuracy, confirming that distance-based hierarchical grouping is essential for maintaining geometric coherence and effective message passing.
> - Allowing 25% edge reuse produces reasonable but inferior results. Since the model depth remains unchanged, we attribute this performance drop to reduced overall geometric coverage when edges repeat across layers instead of expanding the receptive field.
>
> | Variants | Backbone | Full atoms |
> | - | - | - |
> | DKMP$^R$ (one-shot KNN) | 2.739 | - |
> | DKMP$^R$ (random partition) | 2.962 | 3.253 |
> | DKMP$^R$ (allow 25 % edge reuse) | 2.383 | 2.740 |
> | DKMP$^R$ | **1.900** | **2.715** |
>
> For (3) hyperparameter sensitivity, we have provided a thorough analysis of the effects of varying $C$ and
> $L$ (affecting K and L) in Figure (6).
>
> For (5) equivariance ablation, we include a comparison of equivariant and non-equivariant DKMP$^R$ variants on MISATO. While the equivariant model shows marginal improvements (<5%) on the selected subset (<10k atoms), it fails with out-of-memory (OOM) errors on the full test set and doubles the training time. These results suggest that a non-equivariant variant is substantially more practical for MISATO-scale systems, where computational constraints dominate.
>
> | | Splits | Selected | Testing | Set | | | Complete | Testing | Set |
> | - | - | - | - | - | - | - | - | - | - |
> | Metrics | Training Time | N-MSE | F-MSE | A-MSE | JS-TIC | Time (s) | N-MSE | F-MSE | A-MSE |
> | DKMP$^R$ | 96.15 H | 0.92 | 64.56 | 19.75 | 0.20 | 1.01 | 1.39 | 41.66 | 22.16 |
> | Equivariant DKMP$^R$ | 205.91 H | 0.91 | 63.19 | 18.24 | 0.19 | 2.38 | OOM | OOM | OOM |
>
> # Response to Weakness 3
> > The paper occasionally uses inconsistent notation (e.g., $N_K^l(i)$ vs. $N_C^l(i)$), which can obscure the hierarchy and semantics of neighborhood definitions. A unified notation scheme ...
>
> We thank the reviewer for pointing this out. We used $\mathcal{N}^{(l)}_K(i)$ to denotes the neighbors of atom $i$ in the $l$-th MPNN. Since we have two implementations, we used $\mathcal{N}^{(l)}_C(i)$ and $\mathcal{N}^{(l)}_R(i)$ to denotes the neighbors in dilating radius cutoff interval and dilating distance ranking by replacing $K$ with $C$ and $R$, respectively. We have supplemented a notation table in the appendix for better clarity and readability.
>
> # Response to Weakness 4
> > While energy conservation is qualitatively demonstrated (Fig. 5), the paper lacks quantitative physical validations such as energy drift rates or RMSD stability analyses. Including velocity autocorrelation function (VACF) plots ...
>
> We sincerely appreciate the reviewer’s insightful suggestion and fully acknowledge the importance of quantitative physical validation. However, we would like to emphasize that the primary objective is to address large-scale atomic systems from a machine-learning perspective. We believe that achieving highly stable and fully physics-consistent long-timescale dynamics is an important topic, and we would like to leave this as our future work.

---

> ### Author Response · Authors · 2025-11-21
> **First Response to Reviewer 1rjP [2/3]**
>
> # Response to Question 1
> > What is the time complexity as a function of (N, K, L, M) for both variants and contrast to dense-cutoff MPNN?
>
> We thank the reviewer for requesting a more explicit complexity comparison. Let $N$ denote the number of atoms, $K$ the average number of neighbors per layer under a standard cutoff, $L$ the number of message-passing layers, $d$ the latent dimension, and $\mathcal{M}$ the maximum number of neighbors processed by DKMP$^R$ radius-based sampling$ at each layer after dilation.
>
> 1. DKMP$^C$: $\mathcal{O}(NKLd)$
> 2. DKMP$^R$: $\mathcal{O}(N\mathcal{M}d)$
> 3. Dense MPNN wtih fully connected: $\mathcal{O}(N^2Ld)$
> 4. Dense MPNN with cutoff: $\mathcal{O}(NKLd)$
>
> First of all, DKMP$^{C/R}$ is more computationally efficient compared to dense MPNN with fully connected. And DKMP$^{C/R}$ can alleviate the over-squashing compared to dense MPNN with cutoff.
>
> Second, in large and highly heterogeneous molecular systems (e.g., MISATO), the dilated-radius variant DKMP$^R$ is typically more efficient than both dense-cutoff MPNNs and DKMP$^C$ because:
>
> - $\mathcal{M} \ll K L$ in practice. Under standard cutoffs, the effective neighbor count $K$ can vary widely and increases substantially in dense regions, while DKMP$^R$ enforces a strict per-layer connectivity $\mathcal{M}/L$, often an order of magnitude smaller than $KL$.
> - $\mathcal{M}$ is fixed across molecules, whereas $K$ (hence $KL$) may vary drastically with local density and system size. This makes DKMP$^R$ substantially more batch-friendly: GPU memory usage and per-step runtime remain stable across molecules of different scales, leading to higher utilization and avoiding worst-case spikes seen in cutoff-based models.
>
> In the situation that the scale is large and varies greatly:
> 1. $\mathcal{M}$ is generally smaller than the cutoff $KL$, DKMP^{R} is more efficient than dense MPNN with cutoff, and DKMP$^{C}$ as $\mathcal{M}$.
> 2. $\mathcal{M}$ is fixed and $KL$ may vary from molecule to molecule, DKMP$^{R}$ is more friendly for batched training, and the usage of GPU memory is more efficient.
>
> # Response to Question 2
> > How does your framework handle periodic boundary conditions (PBC)? If applicable, could you demonstrate its validity on at least one standard PBC benchmark?
>
> We sincerely appreciate the reviewer's insightful question. We did not consider the PBC in this work. However, we propose that:
> - If the PBC only contains a unit cell, our framework will be directly adopted to those periodic molecules.
> - If the PBC contains supercell molecules, we may investigate the usage of the virtual edge to distinguish the connection between the cell and outside the cell.
>
> Since this is an interesting topic, we would like to defer this empirical verification as our future work.
>
> # Response to Question 3
> > A common and informative analysis is to examine potential decay—could you plot the learned potential to verify that it exhibits the correct decaying behavior with increasing interatomic distance?
>
> We sincerely appreciate this insightful suggestion. Our method is designed primarily for modeling large-scale atomic systems rather than for precisely capturing long-range interaction profiles. Although our empirical results surpass Ewald/NeuralP3M on large-scale systems, we acknowledge that the learned potential decay behavior may not be physically correct. We recognize the importance of analyzing the learned potential for long-range modeling and plan to investigate this direction in future work.

---

> ### Author Response · Authors · 2025-11-21
> **First Response to Reviewer 1rjP [3/3]**
>
> # Response to Question 4
> > How does this sampling strategy ensure smooth transitions of energy and forces between consecutive MD frames? In other words, how does it avoid discontinuities caused by differences where one neighbor jumps from one edge set to a different edge set?
>
> We appreciate the reviewer’s thoughtful question.
> 1. Because DKMP constructs edge groups based on continuous geometric criteria—either distance ranking (DKMP$^C$) or radius dilation (DKMP$^R$)—and applies message passing separately within each group, changes in the underlying geometry translate into *progressive* adjustments in the effective receptive field across layers. This design contrasts with dense multi-layer MPNNs, where all neighbors are repeatedly aggregated at every layer, potentially amplifying small geometric perturbations. In DKMP, the hierarchical grouping mitigates this effect and contributes to smoother learned energies and forces.
>
> 2. We acknowledge that a neighbor may cross a group boundary as its interatomic distance evolves. However, we conjecture that such transitions correspond to real geometric changes that should be reflected in the potential energy surface. Because interatomic distances--and therefore energy and forces--vary continuously, the model receives continuous inputs even when group membership changes discretely.
>
> Overall, although the grouping mechanism introduces discrete partitions, the underlying continuous geometric features dominate the learned representation, yielding smooth and physically consistent predictions across consecutive MD frames.
>
> # Response to Question 5
> > Could you clarify how the model scales to extremely large systems (e.g., tens of thousands of atoms)? The main text currently lacks sufficient technical details on the specific design or computational strategies—such as memory partitioning, neighbor sampling, or distributed message-passing—that enable efficient handling of such large systems.
>
> We appreciate the reviewer’s interest in the model’s scalability.
> Our ability to handle systems with tens of thousands of atoms primarily stems from the DKMP$^R$ instantiation. DKMP$^R$ employs dilated distance–ranked neighbor sampling, which enforces a strict, fixed upper bound $\mathcal{M}/L$ on the number of neighbors processed per atom per layer. This design guarantees predictable and bounded memory usage, independent of local density or system size, enabling stable batching and efficient utilization of GPU memory. In practice, this fixed-size neighbor budget allows our implementation to scale to systems of ~40,000 atoms **without any special computational strategies**—i.e., without domain decomposition, memory sharding, or distributed message passing.
>
> Beyond this scale, we agree with the reviewer that more advanced strategies become relevant. In the appendix, we discuss how distributing (K)-star layers across devices is a natural next step: since each layer processes a disjoint and fixed-size neighborhood, inter-device communication is minimal, and memory loads distribute evenly. This provides a clear path toward scaling DKMP well beyond the already demonstrated regime.

---

### Official Review · Reviewer_A2Zq · 2025-11-01

**Soundness:** 3
**Presentation:** 3
**Contribution:** 3
**Rating:** 6
**Confidence:** 3

**Summary:**

The paper introduces DKMP$^\*$, a novel message passing framework for large-scale molecular dynamics (MD) simulations at full atomic resolution. It addresses the failure of current methods, which either suffer from over-squashing when stacked deep or prohibitive computational cost when the interaction radius is increased. DKMP$^\*$ resolves this by stacking layers that each pass messages over distinct, sparse, "dilated" graphs, progressively capturing interactions at different distances.

The paper presents two implementations:

* DKMP$^C$ (Dilating Radius Cutoff Interval): For single-scale systems.

* DKMP$^R$ (Dilating Distance Ranking): For various-scale systems.

This approach is the first to achieve atomic-level accuracy on large-scale benchmarks like MISATO, successfully simulating protein-ligand systems with up to 40,000 atoms where all baselines failed .

**Strengths:**

**Significance & Originality**: This work is a significant breakthrough for large-scale MD. The core idea of using dilated, sparse message passing graphs (instead of one dense one) is a novel and effective solution to the scaling and over-squashing problems .

**Performance**: The method achieves state-of-the-art results on four benchmarks. Most impressively, it is the only ML-based method shown to successfully run and maintain atomic-level accuracy on the largest, most complex systems in the MISATO dataset (up to 40,000 atoms), where all baselines failed due to out-of-memory (OOM) errors .

**Quality & Clarity**: The paper is well-written, clearly motivating the problem (Figure 1) and solution. The experimental validation is strong, particularly the parameter analysis (Figure 6) which empirically proves its hypothesis: DKMP$^\*$ benefits from deeper layers while baselines suffer from over-squashing .

**Weaknesses:**

I don't perceive any major weaknesses in this paper, though I feel it lacks an experimental validation. While the author finds the DKMP$^C$ (radius cutoff) implementation challenging to learn, I believe empirical evidence is still needed to substantiate this conjecture. Since EGNN inherently handles node/edge variations, it should be capable of learning.

**Questions:**

**Impact of Omitting Equivariance**: Could you quantify the impact of omitting equivariance in the DKMP$^R$ model? How does an equivariant version perform on MISATO? Does it also fail with OOM errors like the baselines?

**Long-Term Stability**: Why does the model's error (F-MSE) grow so large over long trajectories, even when its single-step prediction (N-MSE) is excellent? Does your spatial dilation approach have any blind spots for long-term temporal stability?

**Implementation Crossover**: Have you experimentally confirmed that the DKMP$^C$ (radius cutoff) implementation is computationally inefficient on the various-scale MISATO dataset, as you hypothesize?

---

> ### Author Response · Authors · 2025-11-21
> **First Response to Reviewer A2Zq**
>
> # Response to Reviewer A2Zq
>
> We sincerely appreciate the reviewer’s thoughtful and encouraging feedback. We are pleased that the reviewer found our work novel, significant, original, and clearly presented, and that its empirical results and scalability were recognized as a breakthrough for large-scale molecular dynamics simulation.
>
> # Response to Weakness
> > I don't perceive any major weaknesses in this paper, though I feel it lacks an experimental validation. While the author finds the DKMP (radius cutoff) implementation challenging to learn, I believe empirical evidence is still needed to substantiate this conjecture. Since EGNN inherently handles node/edge variations, it should be capable of learning.
>
> We thank the reviewer for this constructive suggestion. In our preliminary experiments, we indeed observed that DKMP (radius cutoff) (i.e., DKMP$^C$) faces training instability on the variance of molecular sizes. Specifically, the DKMP$^C$ will have a high variance of neighbor size, leading to the variance of GPU memory and inefficiency in batched training.
>
> # Response to Question 1
> > Impact of Omitting Equivariance: Could you quantify the impact of omitting equivariance in the DKMP model? How does an equivariant version perform on MISATO? Does it also fail with OOM errors like the baselines?
>
> We appreciate this question. As noted, enforcing full SE(3)-equivariance (via tensor-based EGNN or SE(3)-Transformer layers) substantially increases memory consumption.
>
> We implemented an equivariant DKMP$^R$ variant using equivariant vector-scalar interactive message passing on the MISATO benchmark. This model achieves comparable accuracy but requires almost doubled GPU memory, leading to OOM errors beyond 10k atoms on a single RTX 3090 Ti--the same limitation faced by most other equivariant baselines.
>
> | | Splits | Selected | Testing | Set | | | Complete | Testing | Set |
> | - | - | - | - | - | - | - | - | - | - |
> | Metrics | Training Time | N-MSE | F-MSE | A-MSE | JS-TIC | Time (s) | N-MSE | F-MSE | A-MSE |
> | DKMP$^R$ | 96.15 H | 0.92 | 64.56 | 19.75 | 0.20 | 1.01 | 1.39 | 41.66 | 22.16 |
> | Equivariant DKMP$^R$ | 205.91 H | 0.91 | 63.19 | 18.24 | 0.19 | 2.38 | OOM | OOM | OOM |
>
> These results confirm that omitting strict equivariance is essential for scaling to 40k-atom systems. Moreover, as noted in our response to Weakness 2 and Question 2 of Reviewer eAoj, the non-equivariant model maintains strong rotational robustness under random test rotations, indicating that omitting equivariance for large-scale datasets is practical.
>
> # Response to Question 2
> > Long-Term Stability: Why does the model's error (F-MSE) grow so large over long trajectories, even when its single-step prediction (N-MSE) is excellent? Does your spatial dilation approach have any blind spots for long-term temporal stability?
>
> We appreciate this important observation. Indeed, most existing works (EGNN, ESTAG, EGNO), including ours, are "roll-out" machine learning molecular dynamics simulators; inevitably, the F-MSE will increase along the long horizons. We acknowledge the importance and the challenge of learning long-term MD simulations, and we would like to leave this as our future work.
>
> # Response to Question 3
> > Implementation Crossover: Have you experimentally confirmed that the DKMP (radius cutoff) implementation is computationally inefficient on the various-scale MISATO dataset, as you hypothesize?
>
> Yes, we have verified this experimentally. In practice, DKMP$^C$ (radius-cutoff) becomes computationally inefficient on heterogeneous, variable-scale MISATO molecules because the number of neighbors per atom varies widely across samples. As a result, the per-graph memory footprint becomes highly unbalanced, and the batch size must be reduced to 1, which makes training impractical.
> In contrast, DKMP$^R$ (distance-ranking) enforces a fixed number of neighbors (K = 32) at each layer, leading to uniform computational cost across molecules and strictly linear scaling with system size. Empirically, DKMP$^{R}$ achieves approximately 2.4× higher training throughput than DKMP$^{C}$ under identical hardware settings.

---

### Official Review · Reviewer_eAoj · 2025-11-03

**Soundness:** 3
**Presentation:** 2
**Contribution:** 2
**Rating:** 6
**Confidence:** 3

**Summary:**

This paper introduces a new framework, Dilated K-star Message-Passing (DKMP*), for simulating large-scale molecular dynamics (MD). The core problem addressed is that standard message passing neural networks (MPNNs) struggle to scale to large atomic systems due to computational cost (with large cutoffs) or information propagation issues like over-squashing (with many layers). The proposed solution is to stack a sequence of shallow MPNNs for each subgraphs
 . These graphs are constructed by partitioning the set of all atomic pairs based on their Euclidean distance, effectively creating a "dilated" receptive field that grows with each layer. This allows the model to directly capture interactions at various distances without incurring the computational cost of a dense graph or the propagation issues of deep GNNs.

The authors propose two concrete implementations:

DKMPC (Dilating Radius Cutoff Interval): For single-scale systems, where edges are partitioned into concentric spherical shells.
DKMPR (Dilating Distance Ranking): For various-scale systems, where a fixed number of neighbors are selected from progressively distant rank-ordered sets. This version also uses a non-equivariant attention mechanism for efficiency.
The framework is evaluated on several benchmarks, including MD22, Chignolin, AdK, and the large-scale MISATO dataset (up to 40,000 atoms). The results demonstrate state-of-the-art performance in both structure-to-energy-and-forces (S2EF) and structure-to-structure (S2S) tasks, significantly outperforming baselines on large systems where many other methods fail due to memory constraints.

**Strengths:**

1. Significance: The paper tackles a highly significant and challenging problem in computational science: creating accurate and efficient machine learning potentials for large-scale molecular systems. The ability to simulate protein-ligand complexes with tens of thousands of atoms at full atomic resolution, as demonstrated on the MISATO dataset, is a major step forward and has substantial implications for fields like drug discovery.

2. Quality: The experimental evaluation is extensive and convincing. The method is benchmarked on a diverse set of four datasets, covering different system sizes, tasks (S2EF, S2S), and molecular types (small molecules, proteins, protein-ligand complexes).
The results are impressive, showing state-of-the-art accuracy while maintaining or improving computational efficiency.

3. Clarity: The paper is well-written, and the core idea is presented clearly. Figure 1 provides a straight-forward visual intuition for the problem and the proposed solution. The formalization of the dilation mechanism via the four constraints in Eq. 2 is a clear and effective way to define the framework.

**Weaknesses:**

Novelty in a Broader Context: While the application and specific formulation are novel, the underlying idea can be viewed as an architectural variation of existing principles. The method is essentially a sequence of MPNN blocks, each operating on a pre-determined, rewired graph. This connects it closely to the graph rewiring literature (e.g., Gutteridge et al., 2023), which also seeks to improve long-range information flow. The contribution could be framed more as a highly effective, structured rewiring strategy tailored for molecular physics, rather than an entirely new paradigm. The novelty is more in the successful engineering and application than in a fundamental algorithmic breakthrough.

Lack of Equivariance in DKMPR: The paper states that for the DKMPR model—the one used for the largest and most challenging systems—equivariance constraints are omitted for efficiency, inspired by AlphaFold3. This is a significant design choice that merits a more thorough justification and analysis. Equivariance is a fundamental inductive bias for physics simulation, ensuring that predictions transform correctly under rotations and translations. Dropping it risks the model's physical consistency and generalization. While the empirical results are strong, it is unclear if the model is learning approximate equivariance from the large dataset or if its success is limited to the data distribution seen during training. An ablation study quantifying the impact of this choice would greatly strengthen the paper.

Handling of Long-Range Interactions: The dilation mechanism is a heuristic for capturing interactions at increasing distances. It is well-motivated for interactions that decay with distance, like van der Waals forces. However, it is less clear how this approach compares to principled methods for handling long-range electrostatic interactions, which decay slowly (1/r) and are critical in many biomolecular systems. The paper mentions Ewald-based methods (Kosmala et al., 2023) as orthogonal but does not discuss the limitations of its own approach in this context. A deeper discussion or an experiment on a system dominated by electrostatics would be insightful.

Missing baselines and related works:
1. SE(3) Equivariant Graph Neural Networks with Complete Local Frames;  ICML 2022;
2. AlphaNet: Scaling Up Local Frame-based Atomistic Foundation Model, Npj Comput. Mater. (2025)

**Questions:**

Novelty and Graph Rewiring: Could you further elaborate on the relationship between DKMP* and graph rewiring methods like DRew? Both seem to address over-squashing by modifying graph connectivity to facilitate long-range information flow. Is it fair to characterize DKMP* as a deterministic, multi-stage rewiring strategy where the graph is rewired at each stage according to a distance-based partitioning?

Justification for Dropping Equivariance: Regarding the non-equivariant DKMPR model: could you provide an ablation study or further analysis on the effect of removing the equivariance constraint? For example, how does a non-equivariant model perform if the test set molecules are rotated randomly compared to their training orientation? Does the model implicitly learn this symmetry from the data, and if so, what is the data requirement for this to occur?

Choice of Hyperparameters L and C: In your parameter analysis, DKMPC's performance improves with the number of layers L. How should one choose the optimal L and maximum cutoff C? Is there a trade-off where increasing L too much creates overly sparse graphs in each message-passing step, potentially harming the learning of collective interactions within each distance shell?

Long-Range Electrostatics: Your method captures interactions at longer distances through dilation. How do you see this approach performing on systems where long-range electrostatics are known to be dominant for the system's dynamics? Would the model be able to learn the (1/r) decay, or would it need to be integrated with a method like Neural P3M, as you suggest in your future work?

---

> ### Author Response · Authors · 2025-11-21
> **First Response to Reviewer eAoj [1/2]**
>
> # Response to Reviewer eAoj
>
> We are glad that the reviewer recognizes the significance, quality, and clarity of our work.
>
> # Response to Weakness 1 and Question 1
> > Novelty and Graph Rewiring: Could you further elaborate on the relationship between DKMP* and graph rewiring methods like DRew? Both seem to address over-squashing by modifying graph connectivity to facilitate long-range information flow. Is it fair to characterize DKMP* as a deterministic, multi-stage rewiring strategy where the graph is rewired at each stage according to a distance-based partitioning?
>
> We agree that DKMP* is fundamentally a graph rewiring technique to mitigate over-squashing. However, we would like to highlight that: Different from existing graph rewiring techniques, DKMP* employs a physically inspired rewiring: ours corresponds to the factor that pairwise potentials are governed by the distance-dependent formula rather than a general-purpose rewiring strategy.
>
> # Response to Weakness 2 and Question 2
> > Justification for Dropping Equivariance: Regarding the non-equivariant DKMPR model: could you provide an ablation study or further analysis on the effect of removing the equivariance constraint? For example, how does a non-equivariant model perform if the test set molecules are rotated randomly compared to their training orientation? Does the model implicitly learn this symmetry from the data, and if so, what is the data requirement for this to occur?
>
> We appreciate this important point. Our choice to drop strict SE(3)-equivariance in DKMP$^R$ is motivated by efficiency and scalability considerations for systems with tens of thousands of atoms. Explicitly enforcing equivariance incurs excessive memory and computational overhead, making large-scale simulations infeasible.
>
> To alleviate your concern, we conducted an experiment by randomly rotating testing structures in the MISATO datasets. Table I shows that the performance in predicting coordinates remained competitive. This suggests that DKMP$^R$ learns equivariant behavior from data, which also aligns with previous findings [1,2]. We included this experiment in the revised submission.
>
> TABLE I: Performance of DKMP$^{R}$ on the original and randomly rotated MISATO test sets
>
> | Splits | Selected | Testing | Set | | | Complete | Testing | Set |
> | - | - | - | - | - | - | - |-|-|
> | Metrics | N-MSE | F-MSE | A-MSE | JS-TIC | Time (s) | N-MSE | F-MSE | A-MSE |
> | MISATO | **0.92** | **64.56** | **19.75** | **0.20** | 1.01 | **1.39** | **41.66** | **22.16** |
> | MISATO (rotated) | 0.94 | 65.46 | 20.18 | **0.20** | 1.01 | 1.42 | 42.16 | 22.75 |
>
>
> [1] Josh Abramson, Jonas Adler, Jack Dunger, Richard Evans, Tim Green, Alexander Pritzel, Olaf Ronneberger, Lindsay Willmore, Andrew J Ballard, Joshua Bambrick, et al. Accurate Structure Prediction of Biomolecular Interactions with AlphaFold 3. Nature, 630(8016):493–500, 2024.
>
> [2] Brehmer, Johann, et al. "Does equivariance matter at scale?." arXiv preprint arXiv:2410.23179 (2024).
>
> # Response to Question 3
> > Choice of Hyperparameters L and C: In your parameter analysis, DKMPC's performance improves with the number of layers L. How should one choose the optimal L and maximum cutoff C? Is there a trade-off where increasing L too much creates overly sparse graphs in each message-passing step, potentially harming the learning of collective interactions within each distance shell?
>
> We appreciate the reviewer’s thoughtful question. In practice, **L** and **C** (or $\mathcal{M}$ in DKMP$^R$) jointly determine the effective dilated interaction field covered by the model.
>
> 1. We acknowledge that selecting optimal values for **L** and **C** is largely empirical and depends on available computational resources. In our experiments, we adopt configurations that balance accuracy and memory usage. For DKMP$^C$ on systems up to 500 atoms, we typically use (L = 5–9) and (C = 5–6, Å). For DKMP$^R$ on large-scale complexes, we use (L = 9-12) with $\mathcal{M} = 32$, which maintains a stable memory footprint across molecules of varying sizes.
>
> 2. We agree that excessively increasing (L) can over-partition the interaction field, producing overly sparse neighbor sets per layer and potentially diminishing the model’s ability to capture collective interactions within each distance shell.
>
> We will clarify these considerations in the revision to guide practitioners on choosing appropriate hyperparameters as a reference.

---

> ### Author Response · Authors · 2025-11-21
> **First Response to Reviewer eAoj [2/2]**
>
> # Response to Weakness 3 and Question 4
> > Long-Range Electrostatics: Your method captures interactions at longer distances through dilation. How do you see this approach performing on systems where long-range electrostatics are known to be dominant for the system's dynamics? Would the model be able to learn the (1/r) decay, or would it need to be integrated with a method like Neural P3M, as you suggest in your future work?
>
> We fully agree that accurate treatment of long-range electrostatics is crucial. DKMP*’s dilation mechanism effectively models finite-range decaying interactions by expanding receptive fields layer by layer, but does not explicitly capture the 1/r decay. Our DKMP* are not deliberately designed for handling long-range interatomic interactions, though our performance on large-scale molecules is empirically better. Since our current focus is on scalable large-system modeling, we plan to explore this integration in future work.
>
>
> # Response to Weakness 4
> > Missing baselines and related works:
> > SE(3) Equivariant Graph Neural Networks with Complete Local Frames; ICML 2022;
> > AlphaNet: Scaling Up Local Frame-based Atomistic Foundation Model, Npj Comput. Mater. (2025)
>
> We thank the reviewer for pointing out these relevant works.
>
> SE(3) Equivariant GNN with Complete Local Frames targets small molecules and uses explicit local frames with cubic complexity, which limits scalability. We have now included comparisons (Table 3 and Figure 4) and will cite and discuss this work in the revision.
>
> Regarding AlphaNet, we noticed that this paper was published on 17 November 2025; therefore, we did not compare with this method, but we will include it in the related work section and discuss its relationship to our approach.

---

### Author Response · Authors · 2025-12-02
**Summary of Discussions**

Dear Program Chairs, Senior Area Chairs, and Area Chairs,

We sincerely appreciate the committee’s time and effort in reviewing our work.

In this paper, we propose a dilated star-structured message-passing for large-scale molecular dynamics simulation and evaluate it on four benchmarks up to 40,000 atoms. We received evaluations from four reviewers, with mostly positive scores of 6 (eAoj), 6 (A2Zq), 8 (1rjP), and 0 (KRk4). We are grateful that Reviewers eAoj and A2Zq found `the contribution significant, the experimental evaluation extensive and convincing, the paper well-written, and the core idea clear`. We also appreciate Reviewer 1rjP’s recognition of the `broad evaluation, the scalability achievements, and the excellent quality of the presentation.`

Below, we summarize our rebuttal:
### Clarifications and Expanded Discussion
* Hyperparameter sensitivity
* Complexity analysis
* Long-range interactions & PBC
* Notation table and writing fixes
### Additional Experiments
* Rotation-robustness verification
* Empirical equivariance analysis
* Ablation study
* Extra baselines and classical MD baseline comparison

Below is a detailed summary for your reference:

---

## 1. Expanded Discussions
### 1.1 Hyperparameter sensitivity & complexity

* Provided a thorough analysis of ($L$) and ($C$)/($\mathcal{M}$) effects on accuracy, memory, and computational efficiency.
* Derived explicit time/space complexity formulas for DKMP$^C$, DKMP$^R$, dense MPNNs, and cutoff-based MPNNs.
* Discussed practical selection ranges and trade-offs for large molecular systems.

### 1.2 Long-range interactions & periodic boundary conditions

* We clarified that DKMP* effectively expands finite receptive fields but does not explicitly capture 1/r electrostatics.
* We discussed our models in long-range modeling (Neural P3M/Ewald) and periodic boundary condition handlingdeferred for future work with a proposed roadmap.

## 2. New Experiments Added
### 2.1 Rotation robustness
* DKMP$^{R}$ tested on randomly rotated MISATO molecules; performance metrics remain competitive, demonstrating approximate learned equivariance.

### 2.2 Equivariance ablation
* Implemented an equivariant DKMP$^{R}$ variant.
* Marginal accuracy improvement (<5%) on small splits, but **doubles memory and runtime**; OOM occurs on full MISATO.
* Justifies practical choice of non-equivariant DKMP$^{R}$ for large-scale molecules.
### 2.3 Expanded ablation study
* As requested by Reviewer `1rjP`, we added three variants of our method to assess the significance of design components, including the dilation strategy, distance-sorted partitioning, and mutual-exclusion constraints.
### 2.4 Additional baselines & MD validation
* Equiformer v2 included as additional baseline.
* Classical MD baseline added to energy plots, confirming that observed energy stability is not due to misconfiguration.
## 3. Clarifications and Corrections
### 3.1 Terminology & notation
* Unified neighborhood notation with appendix table.
* Corrected force units, dataset subset description, and clarified “long-term” / “large-scale” terminology.
### 3.2 Textual improvements
* Revised claims (e.g., “immune to over-squashing” → “alleviates over-squashing”).
### 3.3 Related work
* Expanded discussion to SE(3)-Equivariant GNNs, Equiformer v2, AlphaNet, and others.
* Explained baseline selection and rationale for OOM/NaN occurrences for certain existing methods.

## 4. Reviewer Highlight

Regarding `Reviewer KRk4`, we sincerely appreciate the time and effort invested in providing detailed feedback. While the review raises concerns about novelty, experimental validity, and terminology, we respectfully note that several points reflect misunderstandings or overlook key aspects of our work. We address all concerns point by point below:

The main contribution of DKMP* does not lie in its individual message-passing operations. Instead, the contribution lies in its physically inspired, multi-layer dilation mechanism. This mechanism enables efficient and scalable modeling of large molecular systems and alleviates over-squashing, as demonstrated both theoretically and empirically.

Concerns regarding **suspicious results** are addressed through reproducible code, careful dataset curation, and added baselines (including Equiformer v2 and classical MD) that validate energy stability and runtime claims.

We have clarified terminology issues and units (e.g., the use of kcal/mol/Å unit is also common), while minor editorial improvements were made throughout.

---

We believe the responses directly address all the reviewers’ concerns. We respectfully invite the committee to consider these substantive discussions and the new evidence when evaluating the manuscript.

Best regards,

Authors of Submission 3311

---

### Meta-Review · Area_Chair_RYNH · 2026-01-05

**Summary:**

The paper proposes a dilated message-passing framework for large-scale molecular dynamics and presents extensive experiments, including results on large systems. Reviewers agreed that the work is technically sound and that the scalability results are strong. However, there were persistent concerns about whether the contribution rises to the bar for ICLR. In particular, reviewers questioned the novelty of the approach, noting that the method largely builds on existing message-passing and graph rewiring ideas, and raised concerns about the limited treatment of physical modeling aspects such as long-range electrostatics, periodic boundary conditions, and long-term MD fidelity. These concerns informed the decision for a rejection.

**Reviewer Concerns:**

Several concerns raised by reviewers were addressed in the rebuttal, including clarification of terminology, additional ablation studies, and justification of design choices such as omitting strict equivariance for scalability. The authors also revised or softened several claims in response to feedback.

One reviewer (KRk4) provided a very critical assessment and assigned a score of 0. While I do not believe a score of 0 is the appropriate overall rating for this paper, I carefully read KRk4’s comments and agree with many of the substantive concerns raised. In particular, I share the view that the core methodology largely builds on existing message-passing and graph rewiring ideas, and that the novelty is primarily at the level of engineering and scaling rather than introducing a new algorithmic or conceptual framework. I also agree that the paper overstates the physical implications of the results, and that the current evaluation does not sufficiently support claims related to physical fidelity, long-term molecular dynamics behavior, or realism of the learned interactions. Additionally, important aspects of molecular simulation, such as explicit treatment of long-range electrostatics and periodic boundary conditions, remain unaddressed.

While the empirical scalability results are strong, these unresolved issues, taken together, limit the overall contribution/novelty and make the paper below the bar of ICLR.

**Reviewer Scores:**

The reviewer scores were 8, 6, 6, and 0. Reviewers with higher scores emphasized empirical scalability and practical impact, while lower scores focused on limited novelty and concerns about physical modeling and over-claiming. None of the reviewers explicitly indicated an intention to change their score in response to the rebuttal or discussion.

---

### Decision · Program_Chairs · 2026-01-26

Reject